# Structure-Aware Spectral Sparsification via Uniform Edge Sampling

**Kaiwen He**
Department of Computer Science
Purdue University
he788@purdue.edu

**Petros Drineas**
Department of Computer Science
Purdue University
pdrineas@purdue.edu

**Rajiv Khanna**
Department of Computer Science
Purdue University
rajivak@purdue.edu

## Abstract

Spectral clustering is a fundamental method for graph partitioning, but its reliance on eigenvector computation limits scalability to massive graphs. Classical sparsification methods preserve spectral properties by sampling edges proportionally to their effective resistances, but require expensive preprocessing to estimate these resistances. We study whether uniform edge sampling—a simple, structure-agnostic strategy—can suffice for spectral clustering. Our main result shows that for graphs admitting a well-separated $k$-clustering, characterized by a large structure ratio $\Upsilon(k) = \lambda_{k+1}/\rho_G(k)$, uniform sampling preserves the spectral subspace used for clustering. Specifically, we prove that uniformly sampling $O(\gamma^2 n \log n/\varepsilon^2)$ edges, where $\gamma$ is the Laplacian condition number, yields a sparsifier whose top $(n-k)$-dimensional eigenspace is approximately orthogonal to the cluster indicators. This ensures that the spectral embedding remains faithful, and clustering quality is preserved. Our analysis introduces new resistance bounds for intra-cluster edges, a rank-$(n-k)$ effective resistance formulation, and a matrix Chernoff bound adapted to the dominant eigenspace. These tools allow us to bypass importance sampling entirely. Conceptually, our result connects recent coreset-based clustering theory to spectral sparsification, showing that under strong clusterability, even uniform sampling is structure-aware. This provides the first provable guarantee that uniform edge sampling suffices for structure-preserving spectral clustering.

## 1 Introduction

Spectral clustering is a fundamental approach for discovering community structure in graphs, with applications such as image segmentation [18]. The technique embeds nodes into a low-dimensional space using the eigenvectors of the graph Laplacian, after which standard clustering algorithms (e.g. $k$-means) can be applied to identify latent groups. However, as real-world graphs grow increasingly large—often with millions of nodes and edges—computing even a handful of eigenvectors of the Laplacian becomes computationally prohibitive. This scalability challenge has spurred interest in spectral graph sparsification, which aims to drastically reduce the number of edges while preserving the graph's key spectral (and hence clustering) properties.

Classical results in spectral sparsification show that it is possible to approximate a graph by sampling edges with probabilities proportional to their effective resistances, yielding high-quality spectral sparsifiers that preserve every eigenvalue up to a $(1 \pm \varepsilon)$ factor [19]. Unfortunately, estimating these

39th Conference on Neural Information Processing Systems (NeurIPS 2025).

effective resistances is itself expensive in practice, as it typically requires solving large Laplacian linear systems or constructing specialized data structures (so-called resistance "sketches") for the graph. This overhead can negate the computational gains of sparsification, especially for large graphs. So it is natural to ask whether simpler sampling methods can succeed. Specifically:

*When can uniform edge sampling—without any heavy preprocessing—suffice to preserve the structure needed for spectral clustering?*

Intuitively, if there are coherent clusters in the data, and we want our sampling design to focus on preserving this structure, then standard samplers (e.g. based on effective resistances) for approximating the entire graph would be an overkill. Looking at the extreme case, if there are disconnected but coherent clusters in the data, surely uniform sampling will suffice in preserving the clustering structure. Relaxing this corner case, in presence of 'strong enough' well-separated cluster structure, with sufficiently more intracluster edges than intercluster edges, simple uniform sampling may still suffice. Our focus is to use spectral properties of the graph to identify sufficient conditions for a *strong enough* $k$-cluster structure to allow uniform sampling for sparsification without compromising the said structure. We formalize this through the structure ratio $\Upsilon(k) = \frac{\lambda_{k+1}}{\rho_G(k)}$, where $\lambda_{k+1}$ is the $(k+1)$-th eigenvalue of the normalized Laplacian and $\rho_G(k)$ is the $k$-way expansion (conductance) constant of the graph (i.e. the minimum over all $k$-partitionings of the maximum cluster conductance). Our main result shows that in graphs with large $\Upsilon(k)$, a fraction of uniformly sampled edges suffices to preserve the pertinent spectral structure for clustering:

**Theorem 1.1.** *(Informal restatement of Theorem 4.3) For a graph with $n$ vertices, by subsampling $m = O(\frac{C \cdot n \log n}{\varepsilon^2})$ edges uniformly at random we obtain a sparsified graph whose top $n-k$ eigenspace remains approximately orthogonal to the original cluster indicators:*

$$\left\| \widetilde{\mathbf{V}}_{n-k} \widetilde{\mathbf{V}}_{n-k}^T \mathbf{C} \right\|_F^2 \leq k \left( \frac{1}{\Upsilon(k)} + \frac{\epsilon}{1-\epsilon} \kappa \right).$$

Here, $C$ and $\kappa$ are constants depending on structural and spectral properties of the graph. In the presence of coherent clusters in the graph, $\Upsilon(k)$ is large. For example, under well-clusterability assumption of [17], $\Upsilon(k) = \Omega(k^2)$. Consequently, the bottom-$k$ spectral embedding of the subsampled graph continues to preserve the cluster structure, even though edges were sampled uniformly without any structural information. As a result, spectral clustering applied to the sparsified Laplacian continues to succeed, inheriting the corresponding guarantees of the original graph.

This finding has both practical and theoretical implications. On the practical side, it justifies a simple and scalable preprocessing step for spectral clustering: one can simply sparsify the graph by uniform sampling, without complex computations, and still reliably recover clusters. By eliminating the need for computing leverage scores or resistances, our approach enables spectral clustering to scale to previously infeasible graph sizes with minimal overhead. On the theoretical side, our results contribute to the emerging understanding of how structural assumptions can yield beyond-worst-case guarantees. In particular, it complements recent advances in the coreset and data reduction literature, which show that under clusterability conditions, even uniform random sampling can produce excellent summaries for clustering problems [4, 14]. Here we provide an analogous message for spectral graph clustering: when a graph is inherently well-clusterable, random uniform sampling of edges is powerful enough to preserve its cluster structure.

**Technical Contributions** We present a structure-aware analysis of spectral clustering under uniform edge sampling. Our key insight is that in graphs with well-separated clusters (characterized by a large structure ratio $\Upsilon(k)$), the eigenvectors associated with those clusters are robust to random edge removals. In what follows, we highlight three main contributions of this work:

- *Structure-aware sparsification guarantee.* We prove that uniform sampling can produce a spectral sparsifier that preserves cluster structure under standard clusterability assumptions. In particular, we show that if $\Upsilon(k)$ is sufficiently large, sampling $m = O(n \log n / \varepsilon^2)$ edges uniformly at random is enough to ensure that the relevant spectral subspace for clustering is approximately preserved. Intuitively, our main theorem (Theorem 4.3) guarantees that the sampled graph Laplacian has its top $(n-k)$-dimensional eigenspace nearly orthogonal to the original $k$-dimensional cluster indicator subspace. Equivalently, the cluster indicator vectors remain (up to a small error) in the span of the bottom $k$ eigenvectors of the sparsified Laplacian. This result is notable because it achieves a form of

structure-preserving spectral sparsification without any adaptive weighting, purely through uniform edge selection.

- *Resistance bounds for intra-cluster structure.* To analyze uniform sampling, we derive new bounds on the effective resistances of edges in clustered graphs. We show that within well-defined clusters, every edge has limited spectral influence. Specifically, leveraging the effective condition number and the cluster expansion constant $\rho_G(k)$, we bound the contribution of any intra-cluster edge to the top $(n - k)$ eigen-spectrum of the Laplacian. Symmetrically, we also bound the spectral effect of inter-cluster edges. These results quantify how strong cluster structure (large $\Upsilon(k)$) constrains the 'spectral mass' of edges, and they are crucial in showing that importance-unaware uniform sampling (which does not distinguish between edges) can still preferentially preserve important connections. Notably, these resistance-based bounds are new and may be of independent interest for understanding spectral properties of clustered graphs.

- *Eigenspace matrix Chernoff analysis.* Our analysis introduces a matrix Chernoff concentration argument tailored to the top-(n-k) eigenvector subspace. After sampling edges uniformly, we study the deviation of the sparsified Laplacian from the original Laplacian on the subspace orthogonal to the cluster indicators. By adapting matrix Chernoff bounds to this specific subspace (rather than the entire space), we prove that the sparsifier's Laplacian remains well-behaved (i.e. within a $(1 \pm \varepsilon)$ factor of the original) on all vectors that are orthogonal to the cluster indicator vectors. This step is key to translating our resistance bounds into a global spectral guarantee.

## 2   Related Work

**Clusterable Graphs and Spectral Clustering.** Our work is motivated by the setting of well-clustered graphs, where the graph contains a pronounced $k$-cluster structure. A convenient way to quantify clusterability is via the structure ratio $\Upsilon(k)$ proposed by Peng et al. [17]. They proved a Structure Theorem for such graphs: if $\Upsilon(k) = \Omega(k^2)$ (or stronger bounds in subsequent refinements), then the subspace spanned by the bottom $k$ Laplacian eigenvectors is close to the subspace spanned by the $k$ cluster indicator vectors. Macgregor and Sun [15] showed that one can guarantee spectral clustering's success under much weaker assumptions on $\Upsilon(k)$ that do not scale exorbitantly with $k$. Beyond these spectral analysis results, there have been complementary advances in testing and detecting cluster structure. For example, Czumaj et al. [8] design property testers for cluster structure in bounded-degree graphs, assuming that the graph can be partitioned into well-connected clusters with low conductance cut between clusters.

**Spectral Graph Sparsification.** A series of foundational works has established that any graph can be approximated by a sparse subgraph in a spectral sense. In their seminal result, Spielman [19] introduced an algorithm to produce an $\epsilon$-spectral sparsifier with $O(n \log n / \epsilon^2)$ edges by sampling edges with probabilities proportional to their effective resistances. These sparsification methods, while powerful, rely on non-uniform edge sampling schemes to skew the sample toward "important" edges. In subsequent work, Batson et al. [2] proved the existence of linear-sized spectral sparsifiers (with $O(n/\epsilon)$ edges) using a deterministic constructive method. In general graphs, uniform random sampling of edges without such weights can fail to preserve spectral properties unless the sample size is prohibitively large (e.g., on adversarial graphs with high-degree vertices or weak connectivity structure) – Recent matrix approximation results formalize this limitation: for instance, Cohen et al. [7] study uniform sampling for matrix approximation and show that uniform row sampling can produce effective approximations for certain matrix classes. Critically, their framework uses uniform sampling as an intermediate step to estimate leverage scores, which are then used for subsequent importance sampling to construct the final approximation. Their two-stage approach improves the efficiency of computing importance sampling distributions but fundamentally still relies on resistance-based sampling for final spectral preservation. Our work differs in showing that under structural assumptions (large $\Upsilon(k)$), uniform sampling alone suffices for preserving the spectral subspace relevant to clustering, without any refinement or resistance estimation step. While their work demonstrates that uniform sampling can be a useful computational tool within importance sampling frameworks, our contribution identifies specific graph structures where the importance sampling phase can be eliminated entirely. There have been alternative methods that aim to compute sparsifies that avoid Laplacian graph solvers entirely: Kapralov and Panigrahy [11] obtain a spectral sparsifier by taking a union of random graph spanners. However, their scheme require a non trivial algorithm for computing importance probabilities over the edges. Peng et al. [16] develop local

appoximation algorithms that estimate effective resistance by exploring only a small portion of the graph, via random walks, achieving $\mathcal{O}(\text{poly} \log n/\epsilon)$ with additive $\epsilon$ error for graphs with bounded mixing time.

**Uniform Sampling for Clustering Coresets.** Classical clustering coresets (e.g., for $k$-means or $k$-median) typically rely on importance sampling, but recent work shows that when the data is well-structured, uniform sampling can yield equally effective—and much simpler—coresets. Braverman et al. [4] present a meta-theorem for constructing clustering coresets via uniform sampling: they obtain the first coresets for many constrained clustering problems (such as capacitated or fair clustering) using uniform sampling instead of sensitivity sampling. However, the technical problem and solution are fundamentally different: Braverman et al. [4] sample data points in metric spaces for distance-based clustering objectives, using VC-dimension theory and cluster-balance parameters, while our paper aims to sample graph edges for spectral sparsification, requiring novel $(n-k)$-effective resistances and matrix concentration bounds to preserve Laplacian eigenspaces for the spectral clustering objective. Focusing on the unconstrained $k$-median objective, Huang and Vishnoi [10] investigate the role of balanced clusters. They define a balancedness parameter $\beta \in (0, 1]$ measuring how evenly the points are distributed among optimal clusters. When $\beta$ is not too small (no cluster is overwhelmingly large or tiny), a uniform sample of only $\text{poly}(k, 1/\beta, 1/\epsilon)$ points yields a $(1 + \epsilon)$-approximation for $k$-median, nearly matching the information-theoretic lower bound on sample complexity. In contrast, without assuming balance ($\beta$ close to 1), any coreset construction must effectively inspect $\Omega(1/\beta)$ points in the worst case. Our work is complementary to this line of research: while Braverman et al. and Huang-Vishnoi sample data points in *metric spaces* for clustering objectives like k-means/k-median, we sample graph edges for spectral graph clustering. The mathematical frameworks differ fundamentally: their work relies on metric space geometry and sensitivity analysis, while ours operates in the spectral graph domain using Laplacian eigenspaces and matrix perturbation theory. These results mirror our theme: uniform sampling performs as well as sophisticated importance sampling when each cluster or component of the data is well-conditioned (e.g. size-balanced or with limited "leverage"). Our contribution can be seen as an analogous statement in the graph setting: if the graph's clusters are sufficiently well-separated (large $\Upsilon(k)$ gap), then each edge is roughly equally "important" and hence uniform edge sampling preserves the spectral clustering structure.

## 3  Background

Let $G = (V, E)$ be an undirected graph where $V$ represents the set of vertices and $E$ the set of edges. For weighted graphs, we denote the weight of an edge between vertices $u$ and $v$ as $w(\{u, v\})$. The adjacency matrix $A$ of graph $G$ has entries $A_{ij}$ representing the weight of the edge between vertices $i$ and $j$ (or 1 for unweighted graphs if an edge exists, 0 otherwise). The degree matrix $D$ is a diagonal matrix where $D_{ii} = \sum_j A_{ij}$ represents the sum of weights of all edges incident to vertex $i$. For any subset $S \subset V$, we define $\text{vol}(S) = \sum_{i \in S} D_{ii}$ as the volume of $S$, representing the total weight of edges incident to vertices in $S$. The standard graph Laplacian is defined as $L = D - A$. The normalized Laplacian is given by $\mathcal{L} = I - D^{-1/2}AD^{-1/2}$, where $I$ is the identity matrix. Both matrices are positive semidefinite with eigenvalues ordered as $0 = \lambda_1 \leq \lambda_2 \leq \cdots \leq \lambda_{|V|}$ for connected graphs. For any vector $x \in \mathbb{R}^{|V|}$, the Laplacian quadratic form gives:

$$x^T L x = \sum_{\{u,v\} \in E} w(\{u, v\})(x[u] - x[v])^2$$

This form reveals the connection between the Laplacian and the cut structure of the graph: it sums the weighted squared difference across each edge, so $x^\top L x$ is small if $x$ is nearly constant on connected components (e.g. indicator vectors of clusters). The Laplacian can be factorized as $L = \sum_{(a,b) \in E} w(\{a, b\})L_{(a,b)} = B^T W B$ where $B \in \mathbb{R}^{|E| \times |V|}$ is the edge-incidence matrix and $W$ is a diagonal matrix of edge weights. The normalized Laplacian is defined as $\mathcal{L} = I - D^{-1/2}AD^{-1/2}$. Both $L$ and $\mathcal{L}$ are positive semidefinite (PSD) matrices. We label their eigenvalues as $0 = \lambda_1 \leq \lambda_2 \leq \cdots \leq \lambda_n$.

**Graph Conductance and Expansion Constant.** For a subset $S \subset V$, the conductance (or Cheeger ratio) is defined as:

$$\phi_G(S) = \frac{|E(S, V \setminus S)|}{\text{vol}(S)}$$

where $|E(S, V \setminus S)|$ represents the sum of weights of edges connecting $S$ to its complement, and $\text{vol}(S)$ is the number of edges in $S$. This ratio is small when $S$ has very few edges leaving it compared to the sum of degrees inside $S$. The conductance of the graph is:

$$\phi(G) = \min_{S:\text{vol}(S) \leq \text{vol}(G)/2} \phi_G(S)$$

A small $\phi(G)$ indicates that the graph has a well-defined 'bottleneck' of few edges separating the graph into two clusters. Cheeger's inequality establishes a fundamental relationship between conductance and the second eigenvalue of the normalized Laplacian: $\frac{\lambda_2}{2} \leq \phi(G) \leq \sqrt{2\lambda_2}$. This tells us that the second eigenvalue is small iff the graph contains a sparse cut. For multi-way clustering, this is generalized as:

$$\rho_G(k) = \min_{S_1,\dots,S_k} \max\{\phi_G(S_i) : i = 1,\dots,k\}$$

$\rho(G)$ is small when we can partition the graph into $k$ clusters each having few inter-cluster edges. Higher-order Cheeger-type inequalities [12] extend the two-cluster case to the $k$-th eigenvalue, providing guarantees for $k$-way spectral partitioning: $\frac{\lambda_k}{Ck^2} \leq \rho_G(k) \leq C'\sqrt{\lambda_k \log k}$, where $C$ and $C'$ are constants. A convenient way to quantify how well-separated $k$ clusters are is the structure ratio $\Upsilon(k)$ defined as

$$\Upsilon(k) = \frac{\lambda_{k+1}}{\rho_G(k)}.$$

This ratio compares the $(k+1)$-th eigenvalue (which increases when there is no "$(k+1)$-st cluster") against the $k$-cluster expansion $\rho_G(k)$ (which decreases when clusters are better insulated). Intuitively, large $\Upsilon(k)$ means the first $k$ eigenvalues $\lambda_2,\dots,\lambda_k$ are very small (signifying $k$ good clusters), but $\lambda_{k+1}$ jumps much higher (no $(k+1)$-th cluster), so the clustering structure is pronounced.

**Spectral Clustering.** The Spectral clustering algorithm partitions the graph by computing the bottom $k$ eigenvectors of the normalized Laplacian $\mathcal{L}$ and using them to embed each vertex into $\mathbb{R}^k$. The resulting matrix $F$, whose $i$-th row represents the embedding of vertex $i$, is then clustered using an algorithm like $k$-means. When the graph has a clear $k$-cluster structure (large $\Upsilon(k)$), theory guarantees that these embedded points (the rows of $F$) cluster tightly around $k$ distinct centers corresponding to the true communities.

**Spectral Sparsification.** Spectral sparsification of a graph aims to reduce the number of edges in a graph while preserving its essential structural properties. A spectral sparsifier is a subgraph $\tilde{G}(V, \tilde{E})$ of $G(V, E)$ with reweighted edges that approximates the quadratic form of the Laplacian for all vectors. Specifically, $\tilde{G}$ is an $\epsilon$-spectral sparsifier of $G$ if:

$$(1 - \epsilon)x^T L x \leq x^T \tilde{L} x \leq (1 + \epsilon)x^T L x, \quad \forall x \in \mathbb{R}^n$$

where $L$ and $\tilde{L}$ are the Laplacians of $G$ and $\tilde{G}$, respectively. Preserving the spectrum in this way also preserves the results of spectral clustering: in particular, the important bottom-$k$ eigenspace of $L$ will have a close counterpart in $\tilde{L}$.

**Effective Resistance.** The effective resistance $R_{\text{eff}}(e)$ of an edge $e = (u, v)$ measures the importance of an edge in the graph's connectivity structure. When viewing the graph as an electrical network with edges as resistors, the effective resistance between vertices $u$ and $v$ is:

$$R_{\text{eff}}(u, v) = (\delta_u - \delta_v)^T \mathbf{L}^\dagger (\delta_u - \delta_v)$$

where $L^\dagger$ is the pseudoinverse of the Laplacian and $\delta_x$ is defined as the standard basis vector with value 1 at index $x$ and 0 elsewhere. The leverage score $\tau_e$ of an edge $e = (u, v)$ with weight $w_e$ is defined as: $\tau_e = w_e R_{\text{eff}}(e)$.

## 4   Results

In the context of graph clustering, it is crucial to understand how the eigenvectors of the Laplacian relate to the underlying cluster structure. For graphs that are well-clustered, the subspace spanned

by the bottom $k$ eigenvectors aligns closely with the optimal cluster indicator matrix. This is demonstrated in the following theorem.

**Theorem 4.1** (Structure Theorem [15, 17]). *Let $\{C_1, \ldots, C_k\}$ be a k-way partition of $G$ achieving $\rho_G(k)$, and let $\Upsilon(k) = \lambda_{k+1}/\rho_G(k)$. Assume that $\{\mathbf{v}_i\}_{i=1}^k$ are the first bottom $k$ eigenvectors of matrix $\mathbf{L}$, and $\mathbf{C}_1, \ldots, \mathbf{C}_k \in \mathbb{R}^n$ are the cluster indicator vectors of $\{C_i\}_{i=1}^k$ with proper normalization (i.e. $\mathbf{c}_i = \frac{\mathbf{1}_{C_i}}{\sqrt{|C_i|}}$). Then, the following statements hold:*

1. *For each cluster $C_i$, there exists a linear combination of the eigenvectors $\mathbf{v}_1, ..., \mathbf{v}_k$. $\hat{\mathbf{v}}_i := \sum_{j=1}^k \alpha_j \mathbf{v}_j$ such that $\|\mathbf{c}_i - \hat{\mathbf{v}}_i\|^2 \leq \frac{1}{\Upsilon(k)}$.*

2. *For each eigenvector $\mathbf{v}_i$, there exists a linear combination of the cluster indicator vectors, $\hat{\mathbf{c}}_i := \sum_{j=1}^k \beta_j \mathbf{c}_j$ such that $\sum_{i=1}^k \|\mathbf{v}_i - \hat{\mathbf{c}}_i\|^2 \leq \frac{k}{\Upsilon(k)}$.*

For completeness, we also provide their proof in Section A.1.

### 4.1 Spectral Sparsification for Well-Clustered Graphs

As our first result, we extend the structure theorem to study how the alignment of the cluster indicator vectors and the bottom $k$ eigenvectors of the Laplacian is preserved under sparsification. Intuitively, when the vertices are well-clusterable, we would expect the Laplacian matrix to be decomposable as $\mathbf{L} = \dot{\mathbf{L}} + \mathbf{E}$, where $\dot{\mathbf{L}}$ is block diagonal representing optimal intracluster edge weights, and $\mathbf{E}$ is a small error matrix of intercluster edge weights. If a graph is well-clusterable, this error matrix should have a very small norm. Using Davis-Kahan (Theorem B.2), the eigenvectors of $\dot{\mathbf{L}}$, which correspond precisely to the cluster indicator vectors, and the eigenvectors of $\mathbf{L}$ should be close together. We first explore sparsification through effective resistances which are known to be able to approximate the underlying graph well [19].

**Theorem 4.2** (Sparsification with Structure Preservation). *Let $G = (V, E)$ be a graph with normalized Laplacian matrix $\mathbf{L}$ that satisfies Theorem 4.1. If we sample $O\left(\frac{n \log n}{\epsilon^2}\right)$ edges proportional to their effective resistance to construct a sparsified Laplacian $\tilde{\mathbf{L}}$, then:*

$$\|\tilde{\mathbf{V}}_{n-k}\tilde{\mathbf{V}}_{n-k}^T\mathbf{C}\|_F^2 \leq \frac{1+\epsilon}{1-\epsilon} \cdot \frac{k\rho_G(k)}{\lambda_{k+1}}$$

*where $\tilde{\mathbf{V}}_{n-k}$ is defined as the matrix of top $n - k$ eigenvectors of $\tilde{\mathbf{L}}$.*

This result expectedly follows from the $\epsilon$-approximation of the graph, which preserves the entire spectral information of the graph up to $\epsilon$ error. This theorem immediately suggests that for clustering, it is possible to sparsify the graph via effective resistance sampling while preserving the useful information of the alignment between the bottom k eigenvectors and the cluster indicator vectors.

Interestingly and perhaps a bit surprisingly, for well-clustered graphs, sampling uniformly at random instead of costly effective resistance sampling also preserves the alignment of cluster indicator vectors and the bottom $k$ eigenspace, hinting that sparsification by sampling edges uniformly at random preserves clusterability of the graph.

**Theorem 4.3.** *([Main result] Sparsification via Uniform Sampling with Structure Preservation) Let $G = (V, E)$ be an unweighted graph with Laplacian matrix $\mathbf{L}$ that satisfies Theorem 4.1. Additionally, suppose there exists clusters $\{C_1, ..., C_k\}$ with k-way cut value $\rho_G(k)$. Let $\kappa = \frac{\lambda_n}{\lambda_{k+1}}$ be the rank $n - k$ condition number, and let $\Upsilon(k) = \frac{\lambda_{k+1}}{\rho_G(k)}$ be the clusterability constant. If we uniformly sample $\mathcal{O}\left(\frac{\kappa^2}{(1-k/\Upsilon(k))^2(1-\rho_G(k))^2} \cdot n\log(n)/\epsilon^2\right)$ edges with proper reweighting, we obtain a sparsified Laplacian $\tilde{\mathbf{L}}$ that satisfies*

$$\|\tilde{\mathbf{V}}_{n-k}\tilde{\mathbf{V}}_{n-k}^T\mathbf{C}\|_F^2 \leq k\left(\frac{1}{\Upsilon(k)} + \frac{\epsilon}{1-\epsilon}\kappa\right) \tag{1}$$

*where $\tilde{\mathbf{V}}_{n-k}$ is defined as the matrix of top $n - k$ eigenvectors of $\tilde{\mathbf{L}}$.*

Theorem 4.3 ensures a practical structural guarantee – it implies that for well-clustered graphs sparsification by uniform sampling still retains approximate alignment of the bottom $k$ eigenvectors with the cluster indicator vectors, and so the spectral clustering algorithm when applied to the sparsified Laplacian will recover the underlying clusters.

*Proof idea:* To prove this claim, we now develop a detailed analysis that connects structural properties of well-clustered graphs to spectral stability under uniform sampling. Specifically, we derive new bounds on rank-$(n - k)$ effective resistances in Section 4.2, quantify the distributional proximity between leverage score sampling and uniform sampling, and apply a matrix Chernoff argument tailored to the dominant eigenspace. These efforts culminate in Theorem 4.8, which formally proves that uniform sampling preserves spectral approximation of the top-$(n-k)$ eigenspace of the Laplacian. Finally, we utilize the preservation of the top-$(n - k)$ eigenspace to show that the projection matrices between the original and sparsified Laplacians are close. This results in an additive bound on the squared Frobenius norm of the alignment between the bottom $k$ eigenspace of the sparsified graph. The detailed proof is in the appendix.

## 4.2 Bounding effective resistances

To justify the effectiveness of uniform edge sampling, we must understand how much spectral influence individual edges exert specifically, through their effective resistances. While classical upper bounds depend only on global spectral properties, they fail to exploit the underlying cluster structure. In this section, we derive new resistance bounds tailored to well-clustered graphs, showing that most intra-cluster edges have uniformly low spectral impact when $\Upsilon(k)$ is large.

Define $\delta_u \in \{0, 1\}^{|V|}$ to be a vector that is 1 at index $u$ and 0 everywhere else. Given an edge $\{a, b\} \in E$, the effective resistance is defined as $\langle \delta_a - \delta_b, \mathbf{L}^+(\delta_a - \delta_b) \rangle$. For $\mathbf{L} = \mathbf{V}\boldsymbol{\Sigma}^2\mathbf{V}^T$, the effective resistance can be expressed and bounded in terms of the second eigenvalue:

$$\langle \delta_u - \delta_v, \mathbf{L}^+(\delta_u - \delta_v) \rangle = \sum_{i=2}^{n} \frac{1}{\sigma_i^2}(\delta_a - \delta_b)^T v_i v_i^T (\delta_a - \delta_b) = \sum_{i=2}^{n} \frac{(v_{ia} - v_{ib})^2}{\lambda_i} \leq \frac{2}{\lambda_2}.$$

This bound is quite weak in that it only leverages information regarding the second eigenvalue. Chandra et al. [5] showed that this global bound is tight [5], but there are other more fine-grained bounds under reasonable assumptions (e.g. under a bounded expansion [3]). For our analysis, we must go beyond worst-case bounds too and examine how edge resistances behave under structural assumptions of well-clusterability. We begin by introducing a rank-$(n - k)$ formulation of effective resistance that is more aligned with the subspace relevant to spectral clustering.

### 4.2.1 Bound on Rank n-k Effective Resistance

Prior work on spectral sparsification often relies on effective resistance as an importance score for edge sampling. However, in clustering applications, this heuristic can be misaligned: inter-cluster edges typically have high resistance, increasing their sampling probability—despite being the very edges we hope to downweight or ignore. In contrast, uniform sampling treats all edges equally, and our goal is to show that it naturally favors preserving intra-cluster structure in well-clustered graphs. To formalize this, we focus not on preserving the entire Laplacian spectrum, but specifically the top $(n - k)$ eigenspace, which captures finer intra-cluster variation. We introduce the notion of rank $(n - k)$ effective resistance, tailored to this subspace, and show that for well-separated clusters, the resistance of intra-cluster edges is uniformly small. This will allow us to argue that even uniform sampling suffices to preserve the dominant spectral structure relevant to clustering.

**Definition 4.4** (Rank $n - k$ Effective Resistance). *Given a graph $G = (V, E)$, let $\mathbf{L} = \mathbf{V}\boldsymbol{\Sigma}\mathbf{V}^T$ be the unnormalized Laplacian. Let $\mathbf{L}_{n-k} := \sum_{i=k+1}^{n} \lambda_i \mathbf{v}_i \mathbf{v}_i^T$. The Rank $(n - k)$ effective resistance between vertices $a, b \in V$ is defined as the following*

$$R_{eff}^{n-k}(a, b) := \langle \delta_a - \delta_b, \mathbf{L}_{n-k}^+(\delta_a - \delta_b) \rangle$$

In the seminal paper by Chandra et al. [5], they have the following bounds for effective resistance:

$$\frac{2}{\lambda_2} \geq R_{\text{eff}}(u, v) \geq \frac{1}{n \cdot \lambda_2}.$$

However, for our applications this lower bound is extremely weak as it scales based on the number of vertices. In the following lemma, we show a lower bound on the effective resistance that is purely based on clusterability and the "rank n-k condition number" of the graph.

**Lemma 4.5.** *Let $\{C_1, ..., C_k\}$ be a partition of $G$ achieving $\rho_G(k)$, and let $\Upsilon(k) = \lambda_{k+1}/\rho_G(k)$. Let $L = BB^T$ be the laplacian of the graph and let $\kappa = \frac{\lambda_n}{\lambda_{k+1}}$. Then for any pairs of vertices $\{a, b\}$ within a cluster, the rank $(n - k)$ effective resistance is bounded by*

$$\frac{2}{\lambda_{k+1}} \geq R_{eff}^{n-k}(a, b) \geq \frac{1}{\kappa}(1 - \frac{k}{\Upsilon(k)})\frac{2}{\lambda_{k+1}}$$

Lemma 4.5 provides an improved structure-aware lower bound on the rank-$(n-k)$ effective resistance of intra-cluster edges. It shows that in graphs with large structure ratio $\Upsilon(k)$ with high connectedness (large $\kappa$), these resistances are tightly bounded, reinforcing the intuition that uniform sampling predominantly selects low-resistance edges and thus implicitly preserves intra-cluster connectivity essential for spectral clustering.

To quantify how close leverage score sampling is to uniform sampling, we next bound the number of inter-cluster edges. In well-clustered graphs, it is natural to expect that the fraction of inter-cluster edges is small relative to the total number of edges. This structural property will us to compare the leverage score distribution to the uniform distribution in the analysis that follows.

**Lemma 4.6** (Intercluster Edges). *Let $\{C_1, \ldots, C_k\}$ be a clustering of $G = (V, E)$ that satisfies $\rho_G(k) = \min_{C_1, \ldots, C_k} \max_{i=1, \ldots, k}\{\phi_G(C_i)\}$. Then the number of inter cluster edges is bounded by*

$$|E_{inter}| \leq \rho_G(k) * |E|$$

### 4.3 Matrix Chernoff Proof of Top-$(n - k)$ Approximation

Having established that intra-cluster edges have bounded rank-$(n-k)$ effective resistance (Lemma 4.5) and that inter-cluster edges are proportionally few (Lemma 4.6), we now analyze how these structural properties translate into spectral guarantees for uniform sampling. Our goal in this section is to show that the Laplacian of a uniformly sampled subgraph approximates the original Laplacian well on the top-$(n - k)$ eigenspace. To do this, we first show that under clusterability assumptions, the leverage score distribution is 'close enough' to uniform to allow concentration.

We first start with notation. Let $\tau_e := R_{eff}^{n-k}(e)$ be the rank $n - k$ effective resistance for an edge $e \in E$. Let $p_e := \frac{\tau_e}{\sum_{e \in E} \tau_e}$ the associated probability distribution based on the effective resistances. Let $p^{unif} := 1/|E| = 1/m$ be the uniform distribution.

**Lemma 4.7.** *Let $\{C_1, ..., C_k\}$ be a partition of $G$ achieving $\rho_G(k)$, and let $\Upsilon(k) = O(k^2)$, $\kappa = \frac{\lambda_n}{\lambda_{k+1}}$. Then we have the following relative upper bound on the leverage score probability distribution*

$$\frac{(1 - k/\Upsilon(k)) * (1 - \rho_G(k))}{\kappa} \cdot p^{unif} \leq p_e \leq \frac{\kappa}{(1 - k/\Upsilon(k)) * (1 - \rho_G(k))} \cdot p^{unif}$$

*for all edges $e \in E$.*

Lemma 4.7 enables us to treat uniform sampling as an approximate surrogate for leverage score sampling in the structured setting. With this in place, we now apply a matrix Chernoff bound to show that the Laplacian of a uniformly sampled subgraph preserves the spectral structure of the original Laplacian on the top-$(n - k)$ eigenspace, leading to our main result:

**Theorem 4.8** (Chernoff via Uniform Sampling). *Consider a graph $G = (V, E)$ with laplacian matrix $\mathbf{L} \in \mathbb{R}^{n \times n}$. Suppose that there exists clusters $\{C_1, ..., C_k\}$ that satisfies $\Upsilon(k) = \frac{\lambda_{k+1}}{\rho_G(k)}$. Let $\kappa = \frac{\lambda_n}{\lambda_{k+1}}$ be the restricted rank $n - k$ condition number of $\mathbf{L}$*

*Let $\mathbf{L}_H$ be the Laplacian of a sparsified graph where we sample edges uniformly. Then by uniformly sampling $\mathcal{O}\left(\frac{\kappa^2}{(1-k/\Upsilon(k))^2(1-\rho_G(k))^2} \cdot n \log(n)/\epsilon^2\right)$ edges, we can guarantee*

$$(1 - \epsilon)x^T\mathbf{L}x \leq x^T\mathbf{L}_Hx \leq (1 + \epsilon)x^T\mathbf{L}x$$

*for all $x \in span(v_{k+1}, ..., v_n)$. In other words, we obtain a spectral sparsifier for the dominant $n - k$ of the original Laplacian.*

# 5 Experiments

We empirically validate our theoretical results by comparing uniform edge sampling against effective resistance sampling on synthetic graphs generated by a Stochastic Block Model [1]. We focus on graphs with $k = 4$ clusters with 200 nodes per cluster. To measure the error, we compute the bottom $k = 4$ eigenvectors of the sparsified graph, and we measure the largest principal angle between the bottom 4 eigenvectors with the true cluster indicator vectors (ie $\| \sin \Theta(\tilde{\mathbf{V}}_k, C) \|_\infty$. Smaller angles indicate better preservation of the cluster structure in the spectral embedding. We evaluate both sampling strategies in two settings.

- **Well Clustered Graphs**: Graphs are generated with large intra-edge to inter-edge probability ratio. The probabilities were chosen to highlight uniform edge sampling for low $\gamma$ values (strong clustering structure).
- **Weakly Clustered Graphs**: Graphs are generated with small intra-edge to inter-edge probability ratio. These ratios corresponds to large $\gamma$ values (bad clustering structure).

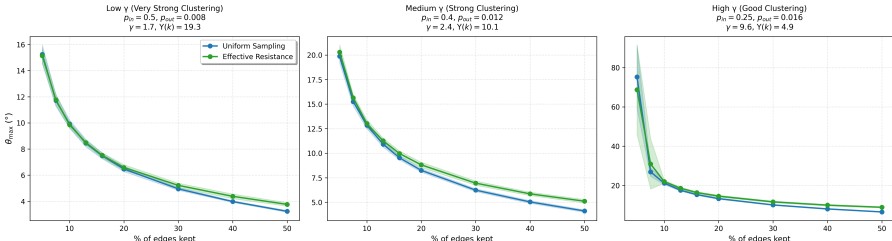

Figure 1: **Good Clusters**: Error plots comparison between Uniform Sampling and Effective Resistance Sampling of strong clusters with varying values of $\gamma$. Shaded region denotes 1 sd over 20 runs.

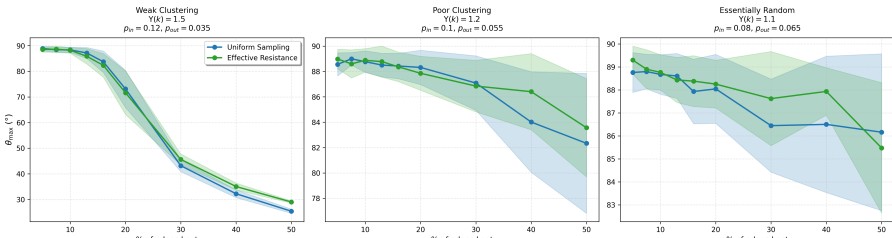

Figure 2: **Poor Clusters**: Error plots comparison between Uniform Sampling and Effective Resistance Sampling of bad clusters with varying values of $\gamma$. Shaded region denotes 1 sd over 20 runs.

For the experiments, the effective resistances were computed via computing the pseudoinverse of the unnormalized Laplacian. All experiments were run on a Macbook Pro M1 with 16GB of RAM. Experiments on the well-clustered graphs confirm the effectiveness of uniform sampling in terms of preserving the bottom-k eigenspace. Surprisingly, even in the poorly-clustered setting, uniform sampling still followed an error trajectory similar to that of effective resistance sampling. Empirically, it is interesting to see that on the well clustered graphs, uniform sampling actually performs slightly better than effective resistance sampling. We hypothesize that this is due to uniform sampling being biased towards *undersampling* cross cluster edges, which results in stronger subspace alignment with the cluster membership vectors. We leave further investigation of this phenomena to future work.

## 5.1 Hierarchical Stochastic Block Model

Experiments are done similarly for a hierarchical stochastic block model. We fix the number of top clusters and sub clusters to be 4 for a total of 16 clusters with the goal of approximating the subspace structure of the top clusters. To test the strength of uniform sampling versus effective resistance sampling, we adjust the probability of a connection between nodes within the same sub cluster ($p_{\text{intra-sub}}$), between nodes of different sub clusters ($p_{\text{inter-sub}}$) and between nodes of different top clusters ($p_{\text{inter-top}}$).

- **Strong Hierarchical Structure**: $p_{\text{intra-sub}} = 0.5$, $p_{\text{inter-sub}} = 0.10$, $p_{\text{inter-top}} = 0.005$
- **Moderate Hierarchical Structure**: $p_{\text{intra-sub}} = 0.35$, $p_{\text{inter-sub}} = 0.08$, $p_{\text{inter-top}} = 0.015$

- **Weak Hierarchical Structure**: $p_{\text{intra-sub}} = 0.20$, $p_{\text{inter-sub}} = 0.06$, $p_{\text{inter-top}} = 0.025$

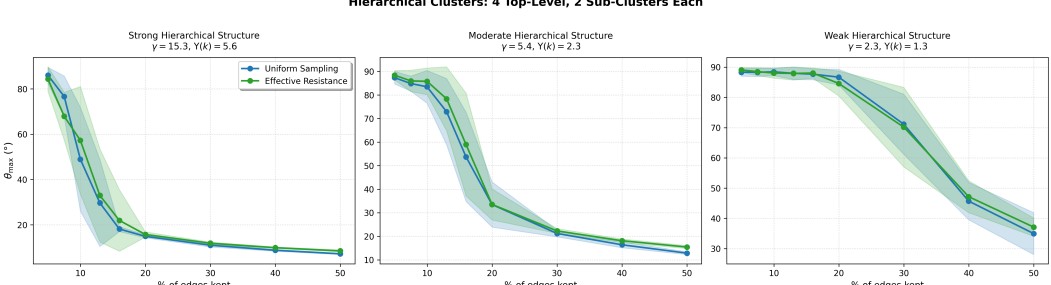

Figure 3: **Hierarchical Clusters**: Error plots comparison between Uniform Sampling and Effective Resistance Sampling of strong clusters with varying values of $\gamma$. Shaded region denotes 1 sd over 20 runs.

## 5.2 Lancichinetti–Fortunato–Radicchi Benchmark Graphs

We perform experiments based on the network benchmark graphs by [13]. Experiments are performed for a network of $800$ nodes. The mixing parameter $\mu$ determines the fraction of edges connecting to others communities, which we vary to generate strong versus weak community structure.

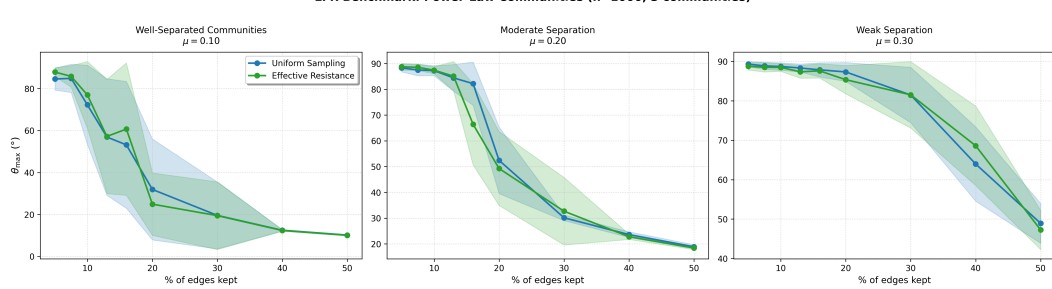

Figure 4: **LFR Network Clusters**: Error plots comparison between Uniform Sampling and Effective Resistance Sampling of strong clusters with varying values of $\mu$. Shaded region denotes 1 sd over 20 runs.

## 6 Conclusion and Future Work

We presented a structure-aware analysis of spectral sparsification via uniform edge sampling, showing that for well-clustered graphs—characterized by a large structure ratio $\Upsilon(k)$—uniform sampling suffices to preserve the spectral subspace critical for clustering. Our approach introduced new resistance bounds for intra-cluster edges, leveraged a rank-$(n-k)$ formulation of effective resistance, and applied a matrix Chernoff analysis to the dominant eigenspace. Together, these tools enabled the first provable guarantees for structure-preserving sparsification using uniform sampling alone.

Several directions remain open for future work. First, our resistance bounds—while sufficient for matrix concentration—may be loose in practice; tightening them, especially by refining the dependence on $\kappa$ and $\Upsilon(k)$, could lead to sharper sampling rates. Second, extending this analysis to weighted graphs or graphs with overlapping clusters may reveal new structural insights. Finally, it would be valuable to explore whether similar structure-aware uniform sampling results can be obtained for other graph problems, such as semi-supervised learning, or spectral embedding beyond clustering.

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

# A  Proofs

## A.1  Proof of Theorem 4.1 from [15]

*Proof of Part 1.* Define $\mathbf{V}_k := [\mathbf{v}_1, ..., \mathbf{v}_i]$. Let vector $\hat{\mathbf{v}}_i$ be the projection of vector $\mathbf{c}_i$ onto the subspace spanned by $\{\mathbf{v}_j\}_{j=1}^k$:

$$\hat{\mathbf{v}}_i := \mathbf{V}_k \mathbf{V}_k^T \mathbf{c}_i = \alpha_1^{(i)} \mathbf{v}_1 + \cdots + \alpha_k^{(i)} \mathbf{v}_k \tag{2}$$

By the definition of Rayleigh quotients:

$$R(\mathbf{c}_i) = \frac{(\alpha_1^{(i)} \mathbf{v}_1 + \cdots + \alpha_n^{(i)} \mathbf{v}_n)^T \mathbf{L}(\alpha_1^{(i)} \mathbf{v}_1 + \cdots + \alpha_n^{(i)} \mathbf{v}_n)}{\|\mathbf{c}_i\|^2} \tag{3}$$

$$= (\alpha_1^{(i)})^2 \lambda_1 + \cdots + (\alpha_n^{(i)})^2 \lambda_n \tag{4}$$

$$\geq (\alpha_2^{(i)})^2 \lambda_2 + \cdots + (\alpha_k^{(i)})^2 \lambda_k + (1 - \alpha' - (\alpha_1^{(i)})^2) \lambda_{k+1} \tag{5}$$

where $\alpha' = (\alpha_2^{(i)})^2 + \cdots + (\alpha_k^{(i)})^2$ and we use the fact that the squared norm of the coefficients of $\mathbf{C}_i$ is 1.

Therefore:

$$1 - \alpha' - (\alpha_1^{(i)})^2 \leq \frac{R(\mathbf{c}_i)}{\lambda_{k+1}} \leq \frac{1}{\Upsilon(k)} \tag{6}$$

And:

$$\|\mathbf{c}_i - \hat{\mathbf{v}}_i\|^2 = (\alpha_{k+1}^{(i)})^2 + \cdots + (\alpha_n^{(i)})^2 = 1 - \alpha' - (\alpha_1^{(i)})^2 \leq \frac{1}{\Upsilon(k)} \tag{7}$$

$\square$

*Proof of Part 2.* Define $\mathbf{C} := [\mathbf{c}_1, \ldots, \mathbf{c}_k]$. For each $1 \leq i \leq k$, let

$$\hat{\mathbf{c}}_i := \mathbf{C}\mathbf{C}^T \mathbf{v}_i = \sum_j \langle \mathbf{v}_i, \mathbf{c}_j \rangle \mathbf{c}_j \tag{8}$$

The objective of bounding $\sum_{i=1}^k \|\mathbf{v}_i - \hat{\mathbf{c}}_i\|^2$ is equivalent to bounding the following projection approximation:

$$\|\mathbf{V}_k \mathbf{V}_k^T \mathbf{C}\|_F^2 \tag{9}$$

Let $\mathbf{L} = \mathbf{V}\Sigma\mathbf{V}^T$ be the eigendecomposition of the Laplacian. Then:

$$\operatorname{tr}(\mathbf{C}^T \mathbf{L}\mathbf{C}) = \operatorname{tr}(\mathbf{C}^T \mathbf{V}\Sigma\mathbf{V}^T \mathbf{C}) \tag{10}$$

$$= \operatorname{tr}(\mathbf{C}^T \mathbf{V}_k \Sigma \mathbf{V}_k^T \mathbf{C}) + \operatorname{tr}(\mathbf{C}^T \mathbf{V}_{n-k} \Sigma \mathbf{V}_{n-k}^T \mathbf{C}) \tag{11}$$

$$\geq \operatorname{tr}(\mathbf{C}^T \mathbf{V}_k \Sigma \mathbf{V}_k^T \mathbf{C}) \tag{12}$$

$$\geq \lambda_{k+1} \operatorname{tr}(\mathbf{C}^T \mathbf{V}_{n-k} \mathbf{V}_{n-k}^T \mathbf{C}) \tag{13}$$

$$= \lambda_{k+1} \|\mathbf{V}_{n-k}^T \mathbf{C}\|_F^2 \tag{14}$$

Note that $\operatorname{tr}(\mathbf{C}^T \mathbf{L}\mathbf{C}) = \sum_{i=1}^k R(\mathbf{C}_i) \leq k\rho_G(k)$. Combining the statements above, we get:

$$\|\mathbf{V}_{n-k}^T \mathbf{C}\|_F^2 \leq \frac{k\rho_G(k)}{\lambda_{k+1}} = \frac{k}{\Upsilon(k)} \tag{15}$$

This gives us the final bound on the approximation of the eigenvectors:

$$k \geq \|\mathbf{V}_k \mathbf{V}_k^T \mathbf{C}\|_F^2 \geq k - \frac{k}{\Upsilon(k)} \tag{16}$$

$\square$

## A.2 Proof of Theorem 4.2

*Proof.* Let $\mathbf{B} = \mathbf{U}\mathbf{\Sigma}^{1/2}\mathbf{V}^T$ be the SVD of the edge-incidence matrix, and $\mathbf{S}$ be a sampling matrix that satisfies $(1 - \epsilon)\mathbf{L} \preceq \tilde{\mathbf{L}} \preceq (1 + \epsilon)\mathbf{L}$ [20]. We have $\mathbf{L} = \mathbf{V}\mathbf{\Sigma}\mathbf{V}^T$ and

$$\tilde{\mathbf{L}} = \mathbf{V}\mathbf{\Sigma}^{1/2}\mathbf{U}^T\mathbf{S}\mathbf{S}^T\mathbf{U}\mathbf{\Sigma}^{1/2}\mathbf{V}^T = \mathbf{L} + \mathbf{V}\mathbf{\Sigma}^{1/2}\mathbf{U}^T\mathbf{E}\mathbf{U}\mathbf{\Sigma}^{1/2}\mathbf{V}^T = \mathbf{L} + \mathbf{\Delta}$$

.

Note that $-\epsilon\mathbf{L} \preceq \mathbf{V}\mathbf{\Sigma}^{1/2}\mathbf{U}^T\mathbf{E}\mathbf{U}\mathbf{\Sigma}^{1/2}\mathbf{V}^T \preceq \epsilon\mathbf{L}$, and since $\mathbf{L}$ is normalized, we have $\|\mathbf{E}\|_2 \leq \epsilon$.

We start with the sum over the Rayleigh quotients of the cluster indicator matrix:

$$\text{tr}(\mathbf{C}^T\mathbf{L}\mathbf{C}) + \text{tr}(\mathbf{C}^T\mathbf{\Delta}\mathbf{C}) = \text{tr}(\mathbf{C}^T\tilde{\mathbf{L}}\mathbf{C})$$

From the eigendecomposition of $\tilde{\mathbf{L}}$, we obtain:

$$k\rho_G(k) + \text{tr}(\mathbf{C}^T\mathbf{\Delta}\mathbf{C}) \geq \tilde{\lambda}_{k+1}\|\tilde{\mathbf{V}}_{n-k}\tilde{\mathbf{V}}_{n-k}^T\mathbf{C}\|_F^2 \tag{17}$$

We can obtain an upper bound on $\text{tr}(\mathbf{C}^T\mathbf{\Delta}\mathbf{C})$:

$$\begin{aligned}
|\text{tr}(\mathbf{C}^T\mathbf{\Delta}\mathbf{C})| &= |\text{tr}(\mathbf{C}^T\mathbf{V}\mathbf{\Sigma}^{1/2}\mathbf{U}^T\mathbf{E}\mathbf{U}\mathbf{\Sigma}^{1/2}\mathbf{V}^T\mathbf{C})| \\
&= |\text{tr}(\mathbf{X}^T\mathbf{E}\mathbf{X})| \\
&= |\text{tr}(\mathbf{X}\mathbf{X}^T\mathbf{E})| \\
&\leq \|\mathbf{X}\mathbf{X}^T\|_F \cdot \|\mathbf{E}\|_2 \\
&\leq \epsilon\|\mathbf{X}\|_F^2 \\
&\leq \epsilon\|\mathbf{U}\mathbf{\Sigma}^{1/2}\mathbf{V}^T\mathbf{C}\|_F^2 \\
&= \epsilon\,\text{tr}(\mathbf{C}^T\mathbf{V}\mathbf{\Sigma}^{1/2}\mathbf{U}^T\mathbf{U}\mathbf{\Sigma}^{1/2}\mathbf{V}^T\mathbf{C}) \\
&= \epsilon\,\text{tr}(\mathbf{C}^T\mathbf{L}\mathbf{C}) = \epsilon \cdot k\rho_G(k)
\end{aligned}$$

Plugging this back into our inequality, we get:

$$\|\tilde{\mathbf{V}}_{n-k}\tilde{\mathbf{V}}_{n-k}^T\mathbf{C}\|_F^2 \leq \frac{(1 + \epsilon)k\rho_G(k)}{\tilde{\lambda}_{k+1}}$$

Using an eigenvalue perturbation result (Theorem B.3), since $\|\mathbf{E}\|_2 < \epsilon$, we have for each $i$:

$$|\lambda_i - \tilde{\lambda}_i| \leq \epsilon\lambda_i$$

This gives us the final bound:

$$\|\tilde{\mathbf{V}}_{n-k}\tilde{\mathbf{V}}_{n-k}^T\mathbf{C}\|_F^2 \leq \frac{(1 + \epsilon)k\rho_G(k)}{(1 - \epsilon)\lambda_{k+1}} = \frac{1 + \epsilon}{1 - \epsilon} \cdot \frac{k}{\Upsilon(k)}$$

$\square$

## A.3 Proof of Theorem 4.3

*Proof.* From Theorem 4.8, we obtain a sparsified graph $H$ with a Laplacian matrix with the following guarantee.

$$(1 - \epsilon)\mathbf{x}^T\mathbf{L}\mathbf{x} \preceq \mathbf{L}_H \preceq (1 + \epsilon)\mathbf{x}^T L\mathbf{x} \quad \forall \mathbf{x} \in \text{span}\{\mathbf{v}_{k+1}, ..., \mathbf{v}_n\}$$

From the structure theorem by Macgregor et al. [15] they have the following bound

$$\|\mathbf{V}_k^T\mathbf{C}\|_F^2 \geq k - \frac{k}{\Upsilon}$$

Now we want to obtain an analogous bound for $\|\tilde{\mathbf{V}}_k^T \mathbf{C}\|_F$, where $\tilde{\mathbf{V}} := [\tilde{\mathbf{V}}_{n-k}, \tilde{\mathbf{V}}_k]$ is the eigenbasis of $\mathbf{L}_H$. We can consider the following

$$
\begin{aligned}
|\|\tilde{\mathbf{V}}_k^T \mathbf{C}\|_F^2 - \|\mathbf{V}_k^T \mathbf{C}\|_F^2| &= |\mathrm{Tr}(\mathbf{C}^T(\tilde{\mathbf{V}}_k \tilde{\mathbf{V}}_k^T - \mathbf{V}_k \mathbf{V}_k^T)\mathbf{C})| \\
&= |\mathrm{Tr}(\mathbf{C}\mathbf{C}^T(\tilde{\mathbf{V}}_k \tilde{\mathbf{V}}_k^T - \mathbf{V}_k \mathbf{V}_k^T))| \\
&\leq \mathrm{Tr}(\mathbf{C}\mathbf{C}^T) \cdot \|\tilde{\mathbf{V}}_k \tilde{\mathbf{V}}_k^T - \mathbf{V}_k \mathbf{V}_k^T\|_2 \\
&= k \cdot \|\tilde{\mathbf{V}}_k \tilde{\mathbf{V}}_k^T - \mathbf{V}_k \mathbf{V}_k^T\|_2
\end{aligned}
$$

This implies that

$$
\begin{aligned}
\|\tilde{\mathbf{V}}_k^T \mathbf{C}\|_F^2 &\geq \|\mathbf{V}_k^T \mathbf{C}\|_F^2 - k \cdot \|\tilde{\mathbf{V}}_k \tilde{\mathbf{V}}_k^T - \mathbf{V}_k \mathbf{V}_k^T\|_2 \\
&\geq k \cdot (1 - \frac{1}{\Upsilon} - \|\tilde{\mathbf{V}}_k \tilde{\mathbf{V}}_k^T - \mathbf{V}_k \mathbf{V}_k^T\|_2)
\end{aligned}
$$

We now proceed to obtain an upper bound for $\|\tilde{\mathbf{V}}_k \tilde{\mathbf{V}}_k^T - \mathbf{V}_k \mathbf{V}_k^T\|_2$

Theorem 4.8 guarantees the following.

$$
(1 - \epsilon)\mathbf{x}^T \mathbf{L}_{n-k} \leq \mathbf{x}^T(\mathbf{L}_{n-k} + \mathbf{E})x \leq (1 + \epsilon)\mathbf{x}^T \mathbf{L}_{n-k}\mathbf{x}
$$

where $\mathbf{E}$ is the error matrix in the dominant $\tilde{\mathbf{V}}_{n-k}$ eigenspace. Note $\|\mathbf{E}\|_2 \leq \epsilon\lambda_n$. Using Theorem B.4, we obtain that

$$
\begin{aligned}
\|\tilde{\mathbf{V}}_{n-k}\tilde{\mathbf{V}}_{n-k}^T - \mathbf{V}_{n-k}\mathbf{V}_{n-k}^T\|_2 &\leq \min(\|\mathbf{L}_{n-k}^+\|_2, (\mathbf{L}_{n-k} + \mathbf{E})^+\|_2) \cdot \|\mathbf{E}\|_2 \\
&\leq \min(1/\lambda_{n-k}, 1/\tilde{\lambda}_{k+1}) \cdot \epsilon\lambda_n \qquad (18) \\
&\leq \epsilon \cdot \frac{\lambda_n}{(1-\epsilon)\lambda_{k+1}}
\end{aligned}
$$

where 18 is from $(1 - \epsilon)\lambda_i \leq \tilde{\lambda}_i \leq (1 + \epsilon)\lambda_i$ for all $i = k+1, ..., n$.

Using this bound we obtain

$$
\|\tilde{\mathbf{V}}_k^T \mathbf{C}\|_F^2 \geq k\left(1 - \frac{1}{\Upsilon} - \frac{\epsilon}{1-\epsilon}\kappa\right) \qquad (19)
$$

$$
\|\tilde{\mathbf{V}}_{n-k}^T \mathbf{C}\|_F^2 \leq \frac{k}{\Upsilon} - \frac{\epsilon}{1-\epsilon}\kappa \qquad (20)
$$

$$
\qquad (21)
$$

$\square$

## A.4    Proof of Lemma 4.5

*Proof.* We start with the upper bound.

$$
\langle \delta_a - \delta_b, \mathbf{L}_{n-k}^+(\delta_a - \delta_b)\rangle = \sum_{i=k+1}^{n} \frac{1}{\lambda_i}(\delta_a - \delta_b)^T v_i v_i^T (\delta_a - \delta_b) \qquad (22)
$$

$$
= \sum_{i=k+1}^{n} \frac{(v_{ia} - v_{ib})^2}{\lambda_i} \qquad (23)
$$

$$
\leq \frac{1}{\lambda_{k+1}}\|V_{n-k}^T(\delta_a - \delta_b)\|_F^2 \qquad (24)
$$

$$
\leq \frac{2}{\lambda_{k+1}} \qquad (25)
$$

To prove the lower bound, we first show that for vertices in the same cluster, the value of the bottom eigenvectors are almost constant. From the second statement of the structure we have that

$$\sum_{i=1}^{k} ||\mathbf{v}_i - \mathbf{C}\mathbf{C}^T\mathbf{v}_i||_\infty^2 \le \sum_{i=1}^{k} ||\mathbf{v}_i - \mathbf{C}\mathbf{C}^T\mathbf{v}_i||_2^2 \le \frac{k}{\Upsilon(k)}$$

$\mathbf{C} = [\mathbf{c}_1, ..., \mathbf{c}_k]$ is defined such that each $c_i$ is the normalized cluster indicator vector. It is important to note that $\mathbf{C}\mathbf{C}^T v_i$ has special structure. the columns of the matrix $\mathbf{C}$ are comprised of constant vectors whose support is defined by the each cluster $C_i$. Hence, when we approximate each $\mathbf{v}_i$ as a linear combination of the columns of $\mathbf{C}$, we get that $\mathbf{C}\mathbf{C}^T v_i$ is a constant k-step function, where each step is defined by each cluster. We define $\bar{v}_i := \mathbf{C}\mathbf{C}^T \mathbf{v}_i$ as the best $k$-step function approximation to $\mathbf{v}_i$. In order to bound the difference between $\mathbf{v}_i(a)$ and $\mathbf{v}_i(b)$, we leverage the fact that vertices $a$ and $b$ are in the same cluster, which implies that

$$\bar{\mathbf{v}}_i(a) = \bar{\mathbf{v}}_i(b)$$

.

Now we can prove the lower bound on the effective resistance

$$R_{\text{eff}}^{n-k}(a, b) = \sum_{i=k+1}^{n} \frac{1}{\lambda_i}(\delta_a - \delta_b)^T \mathbf{v}_i\mathbf{v}_i^T(\delta_a - \delta_b) \tag{26}$$

$$= \sum_{i=k+1}^{n} \frac{(\mathbf{v}_i(a) - \mathbf{v}_i(b))^2}{\lambda_i} \tag{27}$$

$$\ge \frac{1}{\lambda_n} \sum_{i=k+1}^{n} (\mathbf{v}_i(a) - \mathbf{v}_i(b))^2 \tag{28}$$

$$\ge \frac{1}{\lambda_n} \cdot (2 - \frac{2k}{\Upsilon(k)}) \tag{29}$$

$$= \frac{1}{\kappa}(1 - \frac{k}{\Upsilon(k)})\frac{2}{\lambda_{k+1}} \tag{30}$$

$\square$

## A.5 Proof of Lemma 4.6

*Proof.*

$$|E_{\text{inter}}| = \frac{1}{2}\sum_{i=1}^{k} |E(V\backslash C_i, C_i)|$$

$$= \frac{1}{2}\sum_{i=1}^{k} \phi_G(C_i) \cdot \text{Vol}(C_i)$$

$$\le \frac{1}{2}\rho_G(k)\sum_{i=1}^{k} \text{Vol}(C_i)$$

$$= \frac{1}{2}\rho_G(k) \cdot \text{Vol}(G)$$

$$= \rho_G(k) \cdot |E|$$

$\square$

## A.6 Proof of Lemma 4.7

*Proof.* We first prove the upper bound. Recall that $p_e = \frac{\tau_e}{\sum_{e \in E} \tau_e}$. Utilizing Lemma 4.5, the numerator can be upper bounded by $\tau_e \le \frac{2}{\lambda_{k+1}}$. We now obtain a lower bound for the denominator.

Given the clusters $\{C_1, ..., C_k\}$, we can partition the edge set into two disjoint set of intercluster and intracluster edges, $E = E_{\text{intra}} \cup E_{\text{inter}}$, where $E_{\text{intra}} \subset E$ is the set of edges whose vertices lie within the same cluster and $E_{\text{inter}}$ is the set of edges whose vertices lie in two different clusters. The denominator can be lower bounded

$$
\begin{aligned}
\sum_{e \in E} \tau_e &= \sum_{e \in E_{\text{intra}}} \tau_e + \sum_{e \in E_{\text{inter}}} \tau_e \\
&\geq \sum_{e \in E_{\text{intra}}} \tau_e \\
&\overset{\text{Lemma 4.5}}{\geq} \sum_{e \in E_{\text{intra}}} \frac{1}{\kappa}(1 - k/\Upsilon)\frac{2}{\lambda_{k+1}} \\
&= (|E| - |E_{\text{intra}}|) \cdot \frac{1}{\kappa}(1 - k/\Upsilon)\frac{2}{\lambda_{k+1}} \\
&\overset{\text{Lemma 4.6}}{\geq} (m - \rho_G(k) \cdot m) \cdot \frac{1}{\kappa}(1 - k/\Upsilon)\frac{2}{\lambda_{k+1}} \\
&= m \cdot (1 - \rho_G(k)) \cdot \frac{1}{\kappa}(1 - k/\Upsilon)\frac{2}{\lambda_{k+1}}
\end{aligned}
$$

Combining the upper bound of the numerator and the lower bound of the denominator we have the final bound

$$
p_e \leq \frac{\kappa}{(1 - \rho_G(k))(1 - k/\Upsilon)} \cdot \frac{1}{m}
$$

The proof for the lower bound follows similarly. $\qquad\square$

## A.7   Proof of Theorem 4.8

*Proof.* We start with notation. Let $\tau_e := R_{\text{eff}}^{n-k}(e)$ be the rank $n - k$ effective resistance for an edge $e \in E$. Let $p_e := \frac{\tau_e}{\sum_{e \in E} \tau_e}$ the associated probability distribution based on the effective resistances. Let $p^{\text{unif}} := 1/|E|$ be the uniform distribution.

Let $\mathbf{P} \in \mathbb{R}^{n \times (n-k)}$ where

$$
\mathbf{P} = \begin{bmatrix} I_{n-k} \\ 0_{k \times (n-k)} \end{bmatrix}
$$

Given a matrix $\mathbf{A} \in \mathbb{R}^{n \times n}$. The operation $\mathbf{P}^T \mathbf{A} \mathbf{P}$ takes the top $n - k \times n - k$ submatrix of $\mathbf{A}$.

Let

$$
\mathbf{X}_e := \begin{cases} \frac{R}{p^{\text{unif}}} \cdot \mathbf{P}^T (\mathbf{L}_{n-k}^{+/2} \delta_e \delta_e^T \mathbf{L}_{n-k}^{+/2}) \mathbf{P}, & \text{w.p. } p^{\text{unif}}/R \\ 0, & \text{otherwise} \end{cases}
$$

Observe that

$$
\begin{aligned}
\mathbb{E} \sum_{e \in E} \mathbf{X}_e &= \sum_{e \in E} \frac{p^{\text{unif}}}{R} \cdot \mathbf{X}_e \\
&= \mathbf{Q}^T \left[ \mathbf{L}_{n-k}^{+/2} (\sum_{e \in E} \delta_e \delta_e^T) \mathbf{L}_{n-k}^{+/2} \right] \mathbf{Q} \\
&= \mathbf{Q}^T \left[ \mathbf{L}_{n-k}^{+/2} \mathbf{L} \mathbf{L}_{n-k}^{+/2} \right] \mathbf{Q} \\
&= I_{n-k}
\end{aligned}
$$

it immediately follows that the smallest and largest singular value of $\mathbb{E} \sum_{e \in E} \mathbf{X}_e$ are equal to $1$.

$$
\mu_{\min}(\mathbb{E} \sum_{e \in E} \mathbf{X}_e) = \mu_{max}(\mathbb{E} \sum_{e \in E} \mathbf{X}_e) = 1
$$

Now we proceed to bound the norm of $\mathbf{X}_e$

$$\begin{aligned}
||\mathbf{X}_e||_2 &= \frac{R}{p^{\text{unif}}}||\mathbf{P}^T(\mathbf{L}_{n-k}^{+/2}\delta_e\delta_e^T\mathbf{L}_{n-k}^{+/2})\mathbf{P}||_2 \\
&\leq \frac{R}{p^{\text{unif}}}||\mathbf{L}_{n-k}^{+/2}\delta_e\delta_e^T\mathbf{L}_{n-k}^{+/2}||_2 \\
&= \frac{R}{p^{\text{unif}}}(\delta_e^T\mathbf{L}_{n-k}^+\delta_e) \\
&= R \cdot \frac{p_e}{p^{\text{unif}}}
\end{aligned}$$

Using 4.7

For the application of the Chernoff bound we choose $R = \frac{\epsilon^2}{3.5\kappa(1-k/\Upsilon)\ln(n-k)}$. Then

$$||\mathbf{X}_e||_2 \leq \frac{\epsilon^2}{3.5\ln(n)} =: L$$

Then

$$\mathbb{P}\{\lambda_{\min}(\sum\mathbf{X}_e) \leq 1-\epsilon\} \leq (n-k)\exp(-\frac{\epsilon^2}{2L}) \leq (n-k)^{-3/2} \tag{31}$$

$$\mathbb{P}\{\lambda_{\max}(\sum\mathbf{X}_e) \geq (1+\epsilon)\} \leq (n-k)\exp(-\frac{\epsilon^2}{3L}) \leq (n-k)^{-1/6} \tag{32}$$

Thus we preserve the top $n-k$ eigenspace with high probability.

The expected number of edges sampled is

$$\sum_{e\in E} p^{\text{unif}}/R \leq \sum_e \kappa(1-k/\Upsilon)p_e/R \tag{33}$$

$$= \kappa^2(1-k/\Upsilon)^2(n-k)\ln(n-k)/\epsilon^2 \tag{34}$$

This follows from the fact that the sum of the rank $n-k$ leverage scores must add up to $n-k$ (ie $\sum_e p_e = n-k$).

For this sampling to hold we need that $R = \frac{\epsilon^2}{3.5\kappa(1-pk/\Upsilon)\ln(n-k)} < |E|$, otherwise the sampling probabilities for each exceeds 1.

$\square$

# B  Useful theorems

**Theorem B.1** (Matrix Chernoff). *Let $\{\mathbf{X}_k\}$ be a finite sequence of independent, random, Hermetian matrices with common dimension n. Assume that*

$$0 \leq \lambda_{\min}(\mathbf{X}_k) \leq \lambda_{\max}(\mathbf{X}_k) \leq L \quad \forall k$$

*Let $Y := \sum_k \mathbf{X}_k$. Define*

$$\mu_{\min} = \lambda_{\min}(\mathbb{E}\mathbf{Y}) \tag{35}$$
$$\mu_{\max} = \lambda_{\max}(\mathbb{E}\mathbf{Y}) \tag{36}$$
$$\tag{37}$$

*Then,*

$$\mathbb{P}\{\lambda_{\min}(\mathbf{Y}) \leq (1-\epsilon)\mu_{\min}\} \leq n\exp(-\frac{\epsilon^2\mu_{\min}}{2L}) \tag{38}$$

$$\mathbb{P}\{\lambda_{\max}(\mathbf{Y}) \geq (1-\epsilon)\mu_{\max}\} \leq n\exp(-\frac{\epsilon^2\mu_{\max}}{3L}) \tag{39}$$

**Theorem B.2** (Davis-Kahan Theorem from [22]). *Let $\mathbf{A}, \hat{\mathbf{A}} \in \mathbb{R}^{p \times p}$ be symmetric, with eigenvalues $\lambda_1 \geq \ldots \geq \lambda_p$ and $\hat{\lambda}_1 \geq \ldots \geq \hat{\lambda}_p$ respectively. Fix $1 \leq r \leq s \leq p$ and assume that $\min(\lambda_{r-1} - \lambda_r, \lambda_s - \lambda_{s+1}) > 0$, where $\lambda_0 := \infty$ and $\lambda_{p+1} := -\infty$. Let $d := s - r + 1$, and let $\mathbf{V} = (\mathbf{v}_r, \mathbf{v}_{r+1}, \ldots, \mathbf{v}_s) \in \mathbb{R}^{p \times d}$ and $\hat{\mathbf{V}} = (\hat{\mathbf{v}}_r, \hat{\mathbf{v}}_{r+1}, \ldots, \hat{\mathbf{v}}_s) \in \mathbb{R}^{p \times d}$ have orthonormal columns satisfying $\mathbf{A}\mathbf{v}_j = \lambda_j \mathbf{v}_j$ and $\hat{\mathbf{A}}\hat{\mathbf{v}}_j = \hat{\lambda}_j \hat{\mathbf{v}}_j$ for $j = r, r+1, \ldots, s$. Then*

$$\|\sin\Theta(\mathbf{V}, \hat{\mathbf{V}})\|_F \leq \frac{2\min(d^{1/2}\|\hat{A} - \mathbf{A}\|_{op}, \|\hat{\mathbf{A}} - \mathbf{A}\|_F)}{\min(\lambda_{r-1} - \lambda_r, \lambda_s - \lambda_{s+1})}. \tag{40}$$

*Moreover, there exists an orthogonal matrix $\hat{\mathbf{O}} \in \mathbb{R}^{d \times d}$ such that*

$$\|\hat{\mathbf{V}}\hat{\mathbf{O}} - \mathbf{V}\|_F \leq \frac{2^{3/2}\min(d^{1/2}\|\hat{\mathbf{A}} - \mathbf{A}\|_{op}, \|\hat{\mathbf{A}} - \mathbf{A}\|_F)}{\min(\lambda_{r-1} - \lambda_r, \lambda_s - \lambda_{s+1})}. \tag{41}$$

**Theorem B.3** (Theorem 2.3 from [9]). *Let $\mathbf{DAD}$ be a symmetric positive definite matrix such that $\mathbf{D}$ is a diagonal matrix and $\mathbf{A}_{ii} = 1$ for all $i$. Let $\mathbf{DED}$ be a perturbation matrix such that $\|\mathbf{E}\|_2 \leq \lambda_{\min}(\mathbf{A})$. Let $\lambda_i$ be the $i$-th eigenvalue of $\mathbf{DAD}$ and let $\lambda_i'$ be the $i$-th eigenvalue of $\mathbf{D}(\mathbf{A} + \mathbf{E})\mathbf{D}$. Then, for all $i$,*

$$|\lambda_i - \lambda_i'| \leq \frac{\|\mathbf{E}\|_2}{\lambda_{\min}(\mathbf{A})}$$

**Theorem B.4** (Orthogonal Projector Distance from [6][21]). *Let $\mathbf{A} \in \mathbb{R}^{n \times n}$ and let $\mathbf{B} := \mathbf{A} + \mathbf{E} \in \mathbb{R}^{n \times n}$. Let $\mathbf{\Pi}_A, \mathbf{\Pi}_B$ be the projection matrix onto the column space of $\mathbf{A}, \mathbf{B}$ respectively. Then we have the following bound*

$$\|\mathbf{\Pi}_A - \mathbf{\Pi}_B\|_2 \leq \min\{\|\mathbf{A}^+\|_2, \|\mathbf{B}^+\|_2\} \cdot \|\mathbf{E}\|_2$$

