# OpenReview forum: "Structure-Aware Spectral Sparsification via Uniform Edge Sampling"
_NeurIPS.cc/2025/Conference — NeurIPS 2025 poster_

### Official Review · Reviewer_xS39 · 2025-06-29

**Clarity:** 3
**Significance:** 1
**Originality:** 2
**Rating:** 3
**Confidence:** 3

**Summary:**

This paper considers the use of uniform edge sampling for sparsifying a graph with the goal of using the sparsified graph for spectral clustering. While uniform sampling cannot be used for sparsification in general, the authors show that for graphs with well-separated k-clusters, it is possible to use uniform sampling. This is interesting because it provides a guarantee that unfiorm edge sampling suffices for preserving certain graph structural properties.

**Questions:**

(1) Can the authors please respond to my questions about the fairness of the experimental runtime comparison, given that effective resistances needn't be computed exactly for sparsification?
(2) Can the authors address my question about the relative cost of nearly-linear time effective resistance computation vs the cost of spectral clustering the sparsified graph?
(3) Given the conditioning assumptions, I am curious whether the authors believe that there could be a wider range of valid sampling distributions. Given that uniform and effective resistance sampling both work, I'm curious if the analysis could be generalized to show something even stronger. (Note: This is just a question out of my curiosity, and should not be considered a negative point against the paper.)

**Ethical Concerns:**

["NO or VERY MINOR ethics concerns only"]

**Final Justification:**

I would like to clarify that I maintain my score of 3 due to remaining unconvinced about the main claims raised in the rebuttal (as described in the chain above).
- I find that the discussion of prior work on effective resistance is incomplete (the authors should have clearly mentioned that effres only need to be computed to constant accuracy in the original submission, so that all reviewers would have been aware of this fact.) This is a very big oversight, in my respectful opinion.
- I don't think the paper clearly articulates when the results lead to a real improvement in the higher order runtimes. This would be okay, if this were an empirical paper, but this is a theory paper, and the overall runtimes should be made extremely explicit and formal.
- Many of my questions from the original review were not addressed satisfactorily (see the chain below) and in some cases were answered incorrectly (in the rebuttal). Although the authors corrected such statements later, I still feel that my original questions were unfairly dismissed as "out of scope" and not ultimately answered.
- As other reviewers and myself noted, I worry that the authors might be overclaiming the novelty of their work as the first to incorporate uniform sampling / alternatives to effres for sparsification.
- Although this is a theory paper (and I am a theorist myself) I find that the empirical demonstration to effres is incomplete, for the reasons articulated in my rebuttal. This is a minor point, and I would not be concerned about it if I felt that the theoretical contributions were described (1) clearly and (2) completely in the sense of comparing to all relevant context of prior work.

**Limitations:**

Yes

**Paper Formatting Concerns:**

I think the abstract in the pdf is meant to be one paragraph, but in this case the abstract in the pdf is 3 paragraphs.

**Quality:**

2

**Strengths And Weaknesses:**

Strengths:
- the paper studies uniform sampling in a new context, and proves an interesting fact about the power of uniform sampling, and their result could be considered surprising given the precedent of effective resistance sampling for structure-preserving sparsification
- the paper is overall well-written in my opinion
- the result itself seems original, but see concerns below

Weaknesses:
- I am not completely sure how interesting/surprising this result is given the prior work of Braverman et al. '22 on the power of uniform sampling. As a side note, I think the authors do not sufficiently describe this related work.
- The main result, while interesting, seems extremely niche to a specific application. I think the paper would be much stronger if the same analytical techniques were extended to more interesting applications.
- I believe that the original Spielman and Teng paper on sparsification by effective resistance, one only requires a constant-approximation to the effective resistance, which can be computed in nearly-linear time. How does this overhead compare to the cost of computing the eigenvectors? This would be interesting to discuss, under different notions of approximate eigenvector computation.
- Likewise, in the experiments, authors compute the effective resistance by explicitly evaluating the pseudoinverse rather than doing any JL-type approximations. This seems like an unfair comparison. I'd be curious how the error vs edges curves look if one calibrates the effective resistance approximation ratio to be larger.

Minor:
- There are a number of typos including missing references (??) in the submitted draft, but this is a minor point

---

> ### Author Rebuttal · Authors · 2025-07-30
>
> As other reviewers have pointed out, there is a notation confusion regarding the rank-$(n-k)$ condition number $\kappa:= \frac{\lambda_n}{\lambda_{k+1}}$, and the Laplacian Condition number $\frac{\kappa^2}{(1-k/\Upsilon(k))^2(1-\rho_G(k))^2}$. For this response, we refer to the Laplacian condition number as $\gamma$. We apologize for the confusion.
>
> **Relation to Braverman et al**
>
> Our work is complementary but fundamentally different from that of Braverman et al. They sample data points (nodes) for clustering coresets in *metric spaces*, while we sample graph edges for spectral sparsification. This difference in sampling targets reflects deeper distinctions in our mathematical frameworks. Their work with metric space clustering involving distance-based objectives like k-means and k-median, whereas we operate in the domain of spectral graph analysis involving Laplacian eigenspaces and matrix perturbation theory. Correspondingly, our structural assumptions differ: They rely on cluster-balance parameters and capacity constraints, while we depend on the spectral structure ratio $\Upsilon(k) = \lambda_{k+1}/\rho_G(k)$ that captures clusterability in eigenspaces. These fundamental differences require distinct technical machinery. They employ VC-dimension theory and sensitivity analysis for point sampling, while our approach introduces rank-(n-k) effective resistance bounds, eigenspace-specific matrix concentration arguments, and novel connections between leverage score and uniform distributions in clustered graphs. Our novel contributions within this broader paradigm (and un-related to Braverman et al) include the first proof that uniform edge sampling preserves spectral clustering structure; new resistance bounds exploiting graph cluster structure rather than worst-case analysis; and matrix Chernoff analysis restricted to clustering-relevant eigenspaces. We will expand our related work section to properly position our contribution within this emerging understanding of when uniform sampling suffices across different domains.
>
> **Regarding niche scope and techniques**
>
> While spectral clustering itself is a specific technique, it has been a long-standing method in many fundamental machine learning and network analysis domains. We very much agree that our analytical techniques can and should be extended to more interesting applications; however, we want to emphasize that our contributions address a fundamental assumption in spectral sparsification for graphs that has stood since the seminal paper of Srivastava and Spielman [3], which established that importance sampling is necessary for spectral preservation. Our contribution provides the first theoretical proof that the assumption can be broken under structural conditions for an important problem like clustering. This breakthrough has implications beyond clustering for any spectral graph algorithm: in applications like streaming/dynamic settings, or distributed graphs, effective resistance computation is infeasible. The spectral sparsification literature is highly mature precisely because it is fundamental. New theoretical insights in such mature fields are often the most impactful as they challenge established paradigms and thus have tremendous downstream impact. Our work connects to the emerging theme showing that uniform sampling can be surprisingly powerful under structural assumptions, extending this insight to the spectral domain for the first time and changing how we think about when sophisticated sampling is necessary versus when simple methods such as uniform sampling suffice.
>
> **Regarding near linear solvers and cost of spectral clustering**
>
> We acknowledge the remarkable progress in near-linear Laplacian solvers, from the seminal Spielman-Teng [4] work  through recent advances by Peng, Kyng-Sachdeva, and others. These achieve $\tilde{\mathcal{O}}(n)$ complexity using sophisticated techniques like combinatorial preconditioning and multilevel methods. However, these methods are mostly theoretical and many barriers remain that limit their real-world applicability. Specifically, the $|E|$ bottleneck always remains: regardless of how fast individual solver operations become, importance sampling requires computing effective resistance for every edge in the graph. For dense graphs with $|E| = \Theta(n^2)$, this means processing $\Theta(n^2)$ edges, which is an unavoidable quadratic cost that persists even with the most advanced approximation schemes. Additionally, while these solvers are theoretically elegant, they involve complex data structures and numerical conditioning that remain challenging to implement robustly, which explains why most practical systems still rely on other standard sparse methods.
>
> There has been substantial work, for example by Koutis, Miller, Peng [2], that provides near optimal time for approximating leverage scores, but even those methods requires machinery like low stretch spanning trees, which make their implementation highly non-trivial. This has been a large motivating factor in our work where we aimed to provide theoretical guarantees on the effectiveness of uniform sampling, which is trivial to implement. Unlike effective resistance sampling, uniform sampling requires constant preprocessing and achieves spectral guarantees that match importance sampling up to a penalty factor $\gamma$ that depends *only* on the graph's spectral structure, not its size. Our bounds show that for well-clustered graphs, this penalty remains moderate while eliminating all preprocessing overhead.
>
> The reviewer asks how the effective resistance computation cost compares to the eigenvector computation. The key insight is that eigenvector computation is unavoidable - we need it for spectral clustering regardless of our sparsification approach. Effective resistance computation represents additional preprocessing overhead that scales with graph density. Our contribution eliminates this preprocessing entirely. For dense well-clustered graphs, this represents a fundamental improvement: from $\Omega(n^2)$ preprocessing plus eigenvector computation to just eigenvector computation. Even if near-linear resistance computation becomes practical, our result shows that this sophisticated machinery is theoretically unnecessary.
>
> **Regarding JL-type approximate effective resistances and comparisons**
>
> There indeed has been significant progress on improving run-times for evaluating effective resistances, including by using JL approximations. However, these are orthogonal to our contributions. Approaches for approximating effective resistances were originally given by Spielman and Srivastava [3], who used JL projections; however their method still **required** a Spielman-Teng solver [4]. The fundamental $|E| = n^2$ bottleneck persists even with these approximations: for dense graphs, computing effective resistances for all edges requires processing $n^2$ edge pairs, leading to $\mathcal{O}(n^2 \log n/\epsilon^2)$ total complexity regardless of how sophisticated the approximation scheme becomes. More critically, all prior approaches require complex data structure maintenance, random projection computations, and careful numerical conditioning. All this machinery is avoided by uniform sampling. We acknowledge that more sophisticated experimental comparisons using approximate resistances would strengthen our empirical evaluation; however, efficient implementations of Spielman-Teng solvers are not readily available for practical experimental comparison. At the scale of our experiments, we believe that computing the psuedoinverse \textbf{once} and then applying it to all of the edges is the most efficient method of computing the effective resistances.
>
> Our theoretical runtime comparison shows that uniform sampling requires $\mathcal{O}(\gamma^2 n \log n/\epsilon^2)$ edges with O(1) preprocessing per edge, followed by an eigen-computation, while the resistance-based approach requires at least $\mathcal{O}(\log n/\epsilon^2)$ preprocessing per edge plus the same eigen-computation. For well-clustered graphs where $\gamma$ remains moderate and **independent** of $n$, uniform sampling provides a fundamental practical advantage by eliminating the preprocessing bottleneck entirely. While more sophisticated comparisons for fairer experimental setup can indeed be constructed, the purpose of our proof-of-concept experiments is to show effectiveness of uniform sampling in structured graphs; we agree approximations can speed up effective resistance calculations through non-trivial implementations (combined with JL approximations) but a fundamental gap will still remain. We will add this discussion in our paper and thank the reviewer for pointing this out.
>
> > I am curious whether the authors believe that there could be a wider range of valid sampling distributions. Given that uniform and effective resistance sampling both work, I'm curious if the analysis could be generalized to show something even stronger. (Note: This is just a question out of my curiosity, and should not be considered a negative point against the paper.)
>
> This is a good question that was also raised by another reviewer. A very promising direction is to use a uniform subsample in order to compute a coarse approximation of the effective resistance. This technique has been explored in Cohen et al. [1], and could be extended to our structured setting. This approach could potentially be a hybrid between uniform and importance sampling without the need for a fast Laplacian solver.
>
> [1] Michael B. Cohen, Yin Tat Lee, Cameron Musco, Christopher Musco, Richard Peng, and Aaron Sidford. Uniform sampling for matrix approximation.
>
> [2] Ioannis Koutis, Gary L. Miller, and Richard Peng. Approaching optimality for solving sdd systems
>
> [3] Daniel A. Spielman and Nikhil Srivastava. Graph sparsification by effective resistances.
>
> [4] Daniel A. Spielman and Shang-Hua Teng. Spectral sparsification of graphs.

---

> > ### Comment · Reviewer_xS39 · 2025-08-03
> > **Response to rebuttal**
> >
> > I really appreciate the authors for getting back to me!
> >
> > Unfortunately, I have several concerns with content of the rebuttal, however, I am open to further discussion if the authors could clarify a few more concerns. Thanks in advance for your time!
> >
> > 1. It is very important to clarify within the paper itself how the uniform sampling strategy improves upon existing works from a runtime perspective in order for readers and reviewers to contextualize the result. While I know that effective resistance estimation requires an $\Omega(|E|)$ pre-processing in general, the paper describes this in extremely generic terms (see Line 4, Lines 30-36) and does not quantify what they mean by "expensive preprocessing." I feel this should be discussed formally and clearly in the introduction to complete the theoretical story.
> >
> > 2.  The paper does not mention that spectral sparsification only requires computing effective resistances to _constant_ accuracy. This is an extremely important detail to include in the submission. This means that the "expensive pre-processing" required for computing effective resistance is really just an $\tilde{O}(|E|)$ factor, which is basically the cost of inputting or reading the graph.
> >
> > 3. I am very confused by the authors claim in the rebuttal that "the resistance-based approach requires $\log(n)/\epsilon^2$-pre-processing time per edge. I believe this is incorrect, as Spielman-Teng sparsification only requires computing effective resistances to constant accuracy, so this should be only $\log(n)$-time pre-processing per edge. Can the authors please clarify? I think that number of sampled edges depends on the sparsification accuracy (but this is also true of uniform sampling). However, the cost of effectieve resistance estimation itself should be independent of $\epsilon$.
> >
> > 3. In light of 2. and 3. from a runtime perspective, the bound in this paper only improves for the restricted regime where (1) the sample complexity bound in Line 237 is smaller than $\tilde{O}(|E|)$ and (2) the cost of the eigenvector computation is $o(n^2)$. Otherwise, the savings the authors claim would only be lower order relative to all of the remaining contributions. Do the authors have concrete examples where both of these conditions hold in order to ground the discussion? I would like to understand whether the cost savings of uniform sampling are just lower order savings.
> >
> > 4. In their rebuttal, the authors write "our contributions address a fundamental assumption in spectral sparsification for graphs that has stood since the seminal paper of Srivastava and Spielman [3], which established that importance sampling is necessary for spectral preservation." I am skeptical and worry that this type of statement disregard some related literature. For example, this paper (https://arxiv.org/pdf/2406.07521) presents a _deterministic_ sparsification procedure which preserves the spectral density. While this prior work doesn't directly study spectal _clustering_ it does show that effective resistance sampling is not required for preserving certain spectral information and that much simpler strategies can suffice. Thus, I disagree that this paper is for the first time challenging the assumption that importance sampling is necessary. This might be a good reference to include, more generally.
> >
> > 5. The authors claimed that Laplacian system solvers are not numerically stable and do not have practical implementations. I am a bit skeptical of this claim. Indeed, Dan Spielman's webpage contains a link to code for Laplacian system solving (https://github.com/danspielman/Laplacians.jl). Could the authors please explain why this is not useful in experiments?
> >
> > 6. The authors claim that Laplacian system solvers are not practically implementable. However, related literature on Laplacian and SDD system solving specifically cites numerical stability as a strength (e.g. https://www.cs.cmu.edu/~15859n/RelatedWork/KOSZ.pdf.) Additionally, to my knowledge, Laplacian system solvers are not known to have the severe numerical stability issues which arise e.g., in methods like conjugate gradient. Can the authors please provide references explaining why they say "Additionally, while these solvers are theoretically elegant, they involve complex data structures and numerical conditioning that remain challenging to implement robustly."?

---

> > > ### Author Response · Authors · 2025-08-04
> > >
> > > We would like to apologize to the reviewer; upon re-reading the original review and our response, it is clear that our response discussed issues that are well beyond the context of our paper, and the reviewer is justifiably pointing out those issues in their comments. We hope that the reviewer will allow us to discuss our response to their comments in light of the contributions of our paper.
> > >
> > > We view our paper as having two contributions: first, using effective resistances, one can sparsify the input graph (edge-wise) and still get clustering guarantees that are comparable to the clustering guarantees of spectral clustering in the full graph. To the best of our knowledge, this is a novel contribution. The reviewer is absolutely correct that approximate effective resistances (up to a constant factor) suffice for our theoretical guarantees, up to an additional constant loss in the approximation accuracy. We did not discuss this in the paper (an oversight on our end, partly because the use of approximate, relative error, leverage scores and constant-factor approximations are reasonably well understood in the randomized linear algebra community). We will add this discussion to our revision. The reviewer is also correct that in this case, $\epsilon$ is a constant and $O(\log n)$ additional preprocessing time suffices to approximately compute effective resistances, at least in theory. We will add this discussion in our revision, and we will also discuss the additive error approximation algorithms for effective resistances, which seem to be more practical, but insufficient for our purposes, at least from a theoretical perspective.
> > >
> > > Then, the second contribution of our paper is the investigation of the regime in which uniform sampling (which will *always* be more efficient than effective resistance sampling, since it does not need any preprocessing time and does not even need to read the whole graph) might work. In particular, we were motivated by the observation that a well-clustered graph might not necessitate careful sampling, simply because it is already well-clustered, as measured by $\Upsilon(k)$.
> > >
> > > We agree with the reviewer that prior work has also pointed out similar phenomena. The key difference lies in \emph{what spectral properties are preserved and under what assumptions}. Prior approaches aimed to preserve \emph{full spectral similarity} (e.g. the entire spectrum or all-pairs effective resistances) of arbitrary graphs, often requiring innovative alternatives to importance sampling. By contrast, our work restricts attention to well-clustered graphs and targets the subspace spanned by the relevant bottom $k$ Laplacian eigenvectors for clustering. This refined focus allows for a simpler uniform sampling strategy under the additional assumption of clusterability. We apologize for the confusion and will update our introduction to properly credit earlier developments and to further clarify that our “first result” claim is specific to the setting of spectral clustering.
> > >
> > > Regarding regime of importance and run time, consider a graph with $k$ communities, each of size on the order of $n/k$. If each community is relatively dense but the connections between communities are sparse, the original graph may have $|E| = O(n^{1+\alpha})$ for $\alpha>0$. For example, if each $k=O(1)$ clusters is a near-clique, $|E|$ is $\Theta(n^2)$, and our uniform sampling bound calls for only $O(n \log n)$ edges (a tiny fraction of the original) to preserve the cluster structure. Even if clusters are not complete, a graph with $|E| = \Theta(n^{4/3})$ or $n \cdot \text{poly}(\log n)$ edges can still satisfy our structural assumptions (large $\Upsilon(k)$) and admit a sparsifier with $O(n \log n)$ edges, and uniform sampling can only speedup the spectral clustering because computing a eigenvectors of a graph with fewer edges is faster.
> > >
> > > Finally, we conclude by stating that the only connection between our paper and Laplacian eigensolvers is in the context of approximating the effective resistances. Our response criticizing such solvers was unfortunate, actually irrelevant to the context of our paper, and was reflecting comments from the Numerical Linear Algebra community regarding the performance of such solvers in practice compared to highly optimized software such as Trilinos for mesh and grid PDE solvers. No such discussion is or will be included in our paper, since it is irrelevant to our work. The efficiency (from a theoretical and/or practical perspective) of such solvers is well beyond the scope and context of our work, and we are happy to delegate this discussion to other research and experts with a better understanding of this area. In our experiments, we will add a caveat mentioning that there could be methods to speed up the effective resistance calculation that may reduce but not eliminate the gap in time. We apologize for the confusion on a topic that is not relevant to this work.

---

> > > > ### Comment · Reviewer_xS39 · 2025-08-09
> > > > **Thank you**
> > > >
> > > > I thank the authors for their response, and I appreciate the effort to clarify some of the statements made in the initial rebuttal. Nonetheless, I remain unconvinced about the contribution.
> > > >
> > > > I understand that the effective resistance $O(|E|)$ pre-processing cost is avoidable if we replace it with uniform sampling. However, in a _theory_ paper, a runtime improvement only matters if it is a runtime improvement in the _leading order_ term of the computation (otherwise, we would all be optimizing constants and logs). It is still not clear to me whether there is any dent in the leading order term in the computation, or if this is jut saving on lower order factors.
> > > >
> > > > My concern is that if this paper is to be considered a pure theory paper, it should give a formal treatment of how this uniform sampling alternative factors into the overall complexity. However, if the paper is to be considered more of a practical paper, the experiments should be more comprehensive, comparing more thoroughly against alternative approaches (as mentioned in my original response.)
> > > >
> > > > Because of these concerns, and the concerns about the claims I raised in my response to the rebuttal, I am currently inclined to maintain my score. However, I will engage in AC-reviewer discussions and revisit this decision.
> > > >
> > > > I thank the authors for their time and efforts to respond.

---

> ### Author Response · Authors · 2025-08-09
> **Clarification**
>
> We respectfully clarify that our runtime improvement does target the leading-order term of computation by removing a costly preprocessing bottleneck, and we have addressed this in the previous rebuttal. In particular, we have identified a regime where by using uniform edge sampling we eliminate the $\Omega(n^2)$ time needed to estimate effective resistances (or leverage scores) across all $|E|$ edges in a dense graph. This $\Omega(n^2)$ step dominated prior spectral sparsification methods, so bypassing it yields an asymptotic speedup in the highest-order term. Under the well-clustered graph assumption, our sample complexity is $O(n\log n)$. This approach aligns with recent theory emphasizing structure-driven speedups, including those by uniform sampling that we have also cited.

---

### Official Review · Reviewer_51ez · 2025-07-01

**Clarity:** 3
**Significance:** 2
**Originality:** 2
**Rating:** 4
**Confidence:** 4

**Summary:**

This paper considers the problem of sparsifying graphs to preserve the subspace used for clustering graphs clusterable into k pieces using spectral clustering. The paper proves that uniform sampling suffices to preserve this clusterability information provided that suitable number of samples are taken relevant to the number of nodes in the graph, certain measures of the condition number of the graph, and other parameters. Additionally, they perform experiments to corroborate their findings.  To obtain this result they introduce a new notion of effective resistance tailored to the desired approximation and use matrix concentration results tailored to their setting.

**Questions:**

Question / Suggestion: In the abstract it is claimed that the number of uniform samples in the final algorithm scales polynomially with the condition number of the matrix. However, later writing suggests that in fact it scales with a different notion of condition number. I think it is important to clarify which is the case early on. I think for context, it is also important to note that uniform sampling is known to give spectral sparsifiers with a number of samples that scale polynomial with the condition number of the matrix (if appropriately defined). From this perspective it seems that the paper’s improvement can be stated simply in terms of what condition number they depend on. I think it would be great if the paper articulated this a little more clearly.

Question / Suggestion: What are the state-of-the-art runtimes / sample complexities for solving the problems of preserving the bottom subspace and for clustering? Are there overlap in techniques. I think this would be helpful to know and clarify a little more.

Here are some additional detailed comments that may be helpful:
* Page 1: Line 8: I’m not sure that \lambda_{k + 1} and rho_G(k)
* Page 1: Line 14: Perhaps “resistance bounds” should be “effective resistance bounds”
* Page 2: Line 54: I’m not sure that the matrix C was defined. Also, it might be good to use a different symbol given the use of the scalar constant C in the same theorem.
* Page 3: Line 124 – 127: I think one of the key results of that paper is that uniform sampling can be used (through appropriate frameworks to produce sparsifiers); this seems relevant and is not suggested by the summary.
* Page 5: Line 178: I am not sure that “Spectral” should be capitalized in “The Spectral clustering”

**Ethical Concerns:**

["NO or VERY MINOR ethics concerns only"]

**Final Justification:**

My score remains the same for the reasons outlined in the original review. The discussion elevated my view of the contributions paper but not quite enough to change the final numerical value; the concerns about significance from the original review persist. I think the paper would benefit from edits to reflect the discussion with the reviewers, in particular the relationship of the results to the known techniques for spectral sparsification.

**Limitations:**

Yes, with the possible exception of the concerns raised elsewhere in the review.

**Paper Formatting Concerns:**

References seem to be broken in the paper (though this may come from how appendix is provided). This seems like a minor concern, but it seemed best to note.

**Quality:**

3

**Strengths And Weaknesses:**

Strengths: Clusterability problems involving graphs are well-studied problems. New algorithms for graph clustering could have various applications and the spectral method is a natural method to try to improve the efficiency of.  More broadly, this paper considers practically motivated notions of graph approximation and proves that simple methods can obtain these approximations. Along the way the paper introduces a natural notion of edge importances and reasons about them. Altogether, this paper could facilitate future theoretical and practical developments for graph clustering and graph approximation more broadly.

Weaknesses: The number of edges that the paper proves suffices scales near-linearly with the number of vertices and polynomially with measures of the input matrix condition number and the clusterability. It is unclear what the practical settings are where this size isn’t prohibitively expense and substantial reductions in the graph size are obtained. Additionally, there are various methods that have been proposed for the tasks that the paper suggests using graph sparsification for, e.g., subspace computation / approximation and graph clustering. It is unclear if the proposed technique gives end-to-end efficiency gains for these downstream tasks (especially as the methods one might consider may already use sampling). Furthermore, as discussed in the “Questions” section, I think the paper could benefit from a little more comparison and relating of the results to prior work and known approaches.

Finally, I would note that the approach taken by the paper seems straightforward once the problem is established. This not necessarily a strength or a weakness but worth noting. If the paper could more completely articulate where which steps are standard and which are more novel or required novelty, it might elevate the submission.

---

> ### Author Rebuttal · Authors · 2025-07-30
>
> > In the abstract it is claimed that the number of uniform samples in the final algorithm scales polynomially with the condition number of the matrix. However, later writing suggests that in fact it scales with a different notion of condition number. I think it is important to clarify which is the case early on. I think for context, it is also important to note that uniform sampling is known to give spectral sparsifiers with a number of samples that scale polynomial with the condition number of the matrix (if appropriately defined). From this perspective it seems that the paper’s improvement can be stated simply in terms of what condition number they depend on. I think it would be great if the paper articulated this a little more clearly.
>
> We apologize for the confusion in our terminology. The reviewer is correct that $\kappa := \lambda_n/\lambda_{k+1} $, which we refer to as the "rank n-k condition number" of the Laplacian. What we refer to as the "Laplacian Condition number" in the abstract is a condensed form of $\frac{\kappa^2}{(1-k/\Upsilon(k))^2(1-\rho_G(k))^2}$, and we mistakenly used the same notation $\kappa$. We will fix this notational error and clarify it in the paper. In our response, we refer to the Laplacian Condition number as $\gamma$ to prevent confusion.
>
> > The number of edges that the paper proves suffices scales near-linearly with the number of vertices and polynomially with measures of the input matrix condition number and the clusterability. It is unclear what the practical settings are where this size isn’t prohibitively expense and substantial reductions in the graph size are obtained. Additionally, there are various methods that have been proposed for the tasks that the paper suggests using graph sparsification for, e.g., subspace computation / approximation and graph clustering. It is unclear if the proposed technique gives end-to-end efficiency gains for these downstream tasks (especially as the methods one might consider may already use sampling).
>
> The reviewer raises a valid concern regarding the practical scalability. Our sample complexity of $\mathcal{O}(\gamma^2 n \log(n))$ can indeed be large when $\gamma$ is large, and we do not provide theoretical bounds on $\gamma$ in terms of properties of the graph. However, our contribution identifies a key property: **$\gamma$ is structural property independent of the size of the graph**. Our results show that for well-clustered graphs, the difficulty of uniform sampling is captured by $\gamma$, which characterizes the spectral structure of clustering, and not the size of the graph. In our experiments, we observe $\kappa \approx 12$ for well-separated clusters, compared to the $n \approx 1,000$ nodes.
>
> We would like to note that the current state-of-the-art involves effective resistances using near-linear time Laplacian solvers [1,2]. These solvers are, in general, of theoretical interest, with limited practical applications. Additionally, even if a practical Laplacian solver existed, one still needs to compute the effective resistances for all edges, which could quadratically (in the number of edges) for dense graphs. In contrast, our approach requires only $\mathcal{O}(\gamma^2 n \log n/\epsilon^2)$ samples ($\gamma^2$ is a constant that doesn't grow with graph size), plus a trivial preprocessing time for uniform sampling.
>
> > Finally, I would note that the approach taken by the paper seems straightforward once the problem is established. This not necessarily a strength or a weakness but worth noting. If the paper could more completely articulate where which steps are standard and which are more novel or required novelty, it might elevate the submission.
>
> We recognize that the algorithm seems straightforward once the rest of the framework is set. We would like to point out that there are several key insights and technical results in our work that are not straightforward but are required to arrive at the straight-forward algorithm. Previous spectral sparsification literature strongly suggested that importance sampling via effective resistances was *necessary* to preserve spectral structure. The fact that Uniform sampling works seems to heavily contradict conventional intuition. Standard effective resistance measures the importance of an edge with respect to the entire spectrum. Our key insight is that for clustering, standard effective resistance captures more information than necessary for clustering. Thus, we formalize the notion of "rank n-k effective resistance," and prove novel bounds: specifically, we prove that the rank n-k effective resistance behaves nicely under clustering assumptions via Lemma 4.7. Such bounds do not hold for the standard definition of the effective resistance. The tight bounds on the full effective resistance by Chandra et al. [3] are not sufficient and would lead to a parameter $\gamma$ that depends on the graph size. Finally, while matrix Chernoff bounds are a standard tool in the sampling literature, adapting them to our setting required non-trivial modifications.
>
> What are the state-of-the-art runtimes / sample complexities for solving the problems of preserving the bottom subspace and for clustering? Are there overlap in techniques. I think this would be helpful to know and clarify a little more.
>
> As far as we know, one could use variants of power method or Krylov methods to approximate the bottom part of the spectrum of a matrix. However, these methods are too expensive and their convergence properties depend on spectral gaps, etc. Also, implicitly, the nearly linear-time Laplacian solvers [1,2] preserve such subspaces as well as the associated singular values, but these are largely theoretical algorithms involving sophisticated data structures such as multilevel/combinatorial preconditioners and are not practical. Providing explicit running times for such methods is challenging; we will add these discussions and pointers to the literature in our revision. We thank the reviewer for bringing this up as it further highlights the importance of our results.
>
> [1] Ioannis Koutis, Gary L. Miller, and Richard Peng. Approaching optimality for solving sdd systems
>
> [2] Daniel A. Spielman and Shang-Hua Teng. Spectral sparsification of graphs.
>
> [3] Ashok K. Chandra, Prabhakar Raghavan, Walter L. Ruzzo, Roman Smolensky, and Prasoon Tiwari. The electrical resistance of a graph captures its commute and cover times.

---

> > ### Comment · Reviewer_51ez · 2025-08-04
> > **Thank You and Response to Rebuttal**
> >
> > Thank you to the authors for their response. Your response slightly elevates my view of the submission, but not enough to change the score at the moment. Additionally, some of the rebuttal, discussions, and additional reviews raises points of concern. A few points of concern still lingering in light of my review, the previous response, and the broader discussion include the following:
> >
> > * I still think it is important that the paper notes that variants of uniform is sampling are known to give a spectral sparsifiers with a number of samples that depend on the condition number of the paper. I think this fact helps give context for the paper.
> > * For the comment “Previous spectral sparsification literature strongly suggested that importance sampling via effective resistances was necessary to preserve spectral structure.” I should note that “Uniform Sampling for Matrix Approximation” already seems to explain how uniform sampling can capture relevant spectral structure of subgraphs.
> > * The applicability concern I raised involved the near dependence on $n$, the sparsity of graphs in practice, and what end-to-end running times are obtainable. Though I recognize contributions of this paper, I should note that this concern remains.
> > * Additionally, perhaps relevant to the discussion, I should note that there are methods for computing spectral sparsifiers that I believe do not use Laplacian system solvers: “Spectral sparsification via random spanner” by Michael Kapralov and Rina Panigrahy.

---

> > > ### Author Response · Authors · 2025-08-04
> > >
> > > We thank the reviewer for their thoughtful questions and suggestions, which have helped us clarify important aspects of our work.
> > >
> > > 1. We are not sure what the reviewer is referring to here. However, we agree that our introduction should be expanded to clarify our contributions and contextualize our work to the larger body of the uniform sampling literature.
> > >
> > > 2.  Cohen et al. use uniform sampling as an intermediate step to estimate effective resistances for subsequent importance sampling. However, they still fundamentally rely on resistance-based sampling for final spectral preservation. Our contribution shows that under structural assumptions, uniform sampling alone suffices without any refinement or resistance estimation, eliminating importance sampling entirely rather than just making it more efficient. In future work, it would be interesting to investigate whether our structural assumptions and results could improve subsequently improve and/or build upon the work of Cohen et al.
> > >
> > > 3. Regarding the downstream task of computing the bottom $k$ eigenvectors, the use of iterative methods such as Lanczos or Power iterations, would benefit from sparsification.
> > > Such methods directly depend on the speed of performing matrix vector products (number of edges), which gives a runtime of $O(|E|\cdot k)$. The ability to sparsify a graph from $|E|=n^{1+\alpha}$, for $\alpha \in (0,1]$, to $\tilde{O}(n\log n)$, has the potential to provide significant speedup gains downstream. For example,  Macgregor [1] presents a method for spectral clustering, based on the power iteration, whose running time is directly proportional to edge sparsity, and could benefit immediately from our approach (see Algorithm 2 and Theorem 3.1). However, we still emphasize that our work is specifically on the guarantees of uniform sampling for sparsification and spectral preservation for spectral clustering, with less focus on the end to end efficiency gains.
> > >
> > > 4. Thank you for this reference. Approaches involving random spanners and sparsification in the random walk model represent another line of work avoiding Laplacian solvers, though they target general spectral preservation rather than exploiting clustering structure. We will include discussion of these alternative sparsification strategies in our related work to provide better context for our edge sampling approach within the broader effort to simplify spectral sparsification procedures.
> > >
> > > [1] Peter Macgregor. Fast and simple spectral clustering in theory and practice, 2023

---

> > > > ### Comment · Reviewer_51ez · 2025-08-07
> > > > **Response to Author Comment**
> > > >
> > > > Thank you for the follow and discussion of the work. To respond to your points:
> > > > * Thank you for the update. Just to clarify, I was referring to that I think uniform sampling can be shown to yield spectral sparsifiers in certain, seemingly relevant conditions (e.g., bounded degrees and bounded condition number).
> > > > * Thank you for the clarification and the thoughts. I believe Cohen et al. also shows that uniform sampling provides spectral properties of certain subgraphs; this seems relevant even, though indeed there are differences between the works.
> > > > * Thank you for clarifying. The concerns I raised remain, with the understanding that they are potentially outside the scope of the submission.
> > > > * Thank you for your reply. To clarify, one of the reasons I provided the reference as the comment as the discussion with another author seemed to potentially suggest that approximations like low stretch spanning trees were need compute the information for sampling sparsifiers; this paper is a potential counterpoint.

---

> > > > > ### Author Response · Authors · 2025-08-07
> > > > >
> > > > > We thank the reviewer for their helpful suggestions and discussions! Regarding other conditions under which uniform sampling yields spectral sparsifiers (e.g., bounded degrees and bounded condition numbers), we agree and it would be interesting to investigate these connections further. Regarding Cohen et al., we agree that there are relevant similarities between our papers, and we will make sure to expand on this comparison in our introduction to better contextualize our contributions within the existing literature on uniform sampling for spectral applications.

---

> > > > > > ### Comment · Reviewer_51ez · 2025-08-08
> > > > > > **Response to Authors**
> > > > > >
> > > > > > Thank you to the authors for the reply. That addresses my lingering questions.

---

### Official Review · Reviewer_R6LN · 2025-07-02

**Clarity:** 2
**Significance:** 3
**Originality:** 3
**Rating:** 4
**Confidence:** 4

**Summary:**

The authors work on the problem of spectral sparsification in the context of graph clustering. their primary contribution is demonstrating that if a graph has a large structure ratio, i.e., it naturally forms k clusters, with few edges in between each cluster, then uniform sampling will produce a sparsifier preserving the top $n-k$ eigenspace of the Laplacian.


The approach by the authors is much more efficient than the usual sparsification approach, which requires computing the leverage scores of the edges, and sampling proportional to them. Their work is more similar in spirit to CLMMPS14, exploring what can be achieved with uniform edge sampling.

**Questions:**

in the abstract $\kappa$ is said to be the laplacian condition number, but i believe it is the ratio of the nth eigenvalue to the k+1th. Is this value polylog in the case of graphs that have large structure ratio?

As someone who is unfamiliar with this notion of the cluster ratio, what do graphs with large cluster ratio look like? If a graph has cluster ratio k^2, how many total edges are there across cuts?

**Ethical Concerns:**

["NO or VERY MINOR ethics concerns only"]

**Final Justification:**

Overall, I would lean towards acceptance.

I feel like this paper addresses a somewhat interesting question in this regime, where we ask what can be achieved with uniform sampling over leverage score sampling. The authors do indeed make an interesting contribution here, by describing precisely a metric, as well as a class of graphs, for which spectral properties are preserved.

I do not personally similarly to the other reviewers about runtime objections here, it seems pretty clear to me that this will run in $O(|V|)$ time. The main reason why I did not give a higher score, is because it does not seem obvious to me what the applications of this work would be. Nevertheless, structural results like these do oftentimes end up finding use eventually somwhere.

**Limitations:**

yes

**Paper Formatting Concerns:**

what is C in theorem 4.3? Is this the diagonal matrix with the cluster weights?

broken reference on line 210, 217

**Quality:**

3

**Strengths And Weaknesses:**

Strengths

The authors prove some nice bounds on what uniform sampling can achieve, and give a concrete condition, large structure ratio, characterising when uniform sampling would perform well. On the way to proving these, they define some new notions for effective resistance, as well as clustering.

Weaknesses

I'm not quite sure how restrictive this graph class is, and what graphs with large structure ratio look like. Furthermore, it is not super clear what the purpose of proving this type of spectral bound is. If the goal is to sparsify graphs that naturally break into components, would running any graph clustering algorithm, then uniformly sampling the individual clusters not also achieve the same spectral bound?

---

> ### Author Rebuttal · Authors · 2025-07-30
>
> > In the abstract $\kappa$ is said to be the Laplacian condition number, but i believe it is the ratio of the nth eigenvalue to the k+1th.}
>
> We apologize for the confusion in our terminology. The reviewer is correct that $\kappa := \lambda_n/\lambda_{k+1} $, which we refer to as the "rank n-k condition number" of the Laplacian. What we refer to as the ``Laplacian Condition number" in the abstract is a condensed form of $\frac{\kappa^2}{(1-k/\Upsilon(k))^2(1-\rho_G(k))^2}$, which we mistakenly used the same notation $\kappa$. We will fix this notation error and clarify it in the paper. In our response, we refer to the Laplacian Condition number as $\gamma$ to prevent confusion.
>
> > I'm not quite sure how restrictive this graph class is, and what graphs with large structure ratio look like.
>
> The reviewer is raising a good point. Indeed, we cannot answer this question in full generality. We do know that graphs whose adjacency matrices have block-like structure with a few number of entries outside the block structure exhibit a large structure ratio. Graphs models like the Stochastic Block Model (SBM), Degree-Adjusted SBMs, and also Hierarchical Random Graphs tend to exhibit larger structure ratio $\Upsilon(k)$.
>
> > it is not super clear what the purpose of proving this type of spectral bound is
>
> The purpose of our spectral bound is to address a fundamental challenge in spectral clustering: it requires computing the bottom k eigenvectors of the Laplacian, but this becomes prohibitively expensive on large graphs. Our bound establishes that uniform sampling of the edges (which corresponds to sparsification of the underlying adjacency matrix) can preserve the clustering-relevant eigenspace (the bottom k eigenvectors) with $\mathcal{O}(\gamma^2 n \log n /\epsilon^2)$ samples, eliminating the need for the rather expensive effective resistance computation entirely. Importantly, the penalty for using uniform sampling (captured by $\gamma$) is *independent* of graph size, meaning the difficulty of uniform sampling is determined by spectral structure, not scale. This provides the first theoretical result of its kind, showing that simple uniform sampling suffices to make spectral clustering scalable while preserving the essential spectral subspace.
>
> > If the goal is to sparsify graphs that naturally break into components, would running any graph clustering algorithm, then uniformly sampling the individual clusters not also achieve the same spectral bound?
>
> This is a great question: The reviewer essentially comments that clustering followed by sparsification might achieve the same goals and bounds. However, the challenge of spectral clustering is to discover the unknown cluster structure of the graph, and spectral clustering becomes prohibitively expensive as graphs get larger. This work highlights that we can use uniform sampling first to preserve the relevant spectral properties of the graph for clustering and, therefore, make spectral clustering cheaper. Existing methods all require a preprocessing step of computing the effective resistances (or approximate effective resistances) for all edges to determine which edges are important for preserving spectral structure. Our contributions show that under clusterability conditions, the expensive preprocessing step is not required - simple uniform sampling suffices.
>
>
> > Is this [$\kappa$] value polylog in the case of graphs that have a large structure ratio?
>
> For graphs with a large structure ratio, the value of $\kappa$ can greatly vary depending on the spectral structure of each individual cluster, and can in fact be a constant with **no dependence** on the size of the graph. For example, if we have two complete graphs with $n$ vertices and connect the two complete graphs with a few edges. The structure of the two complete graphs dominate the eigenspace from $\lambda_2$ to $\lambda_{2n}$, which results in $\kappa = \mathcal{O}(1)$.
>
> > As someone who is unfamiliar with this notion of the cluster ratio, what do graphs with large cluster ratio look like? If a graph has cluster ratio $k^2$, how many total edges are there across cuts?
>
> From an intuitive perspective, the cluster measure ratio of the strength of each cluster (captured by $\lambda_{k+1})$ to the proportion of edge crossings between clusters (captured by $\rho_G(k)$). From the example of the SBM, the $p_{in}$ value determines the strength of individual clusters, which should increase $\lambda_{k+1}$, while a low value of $p_{out}$ corresponds to a smaller ratio of inter-cluster edges to intra-cluster edges, decreasing the value of $\rho_G(k)$.
>
> If a graph has cluster ratio $\Upsilon(k) = k^2$, this gives us the $k$-way cut cost of $\rho_G(k) = \frac{\lambda_{k+1}}{k^2}$. From Lemma 4.3, in the paper, we have the bound on the number of inter-cluster edges $|E_{inter}| \leq \rho_G(k)\cdot |E|$. This gives an upper bound of $|E|\lambda_{k+1}/k^2$.

---

> > ### Comment · Reviewer_R6LN · 2025-08-07
> >
> > Thank you for the response, this was a very interesting read.
> >
> > Reading through the other reviewers comments, I do mostly agree with the authors on their points. This may be coming from a theoretical perspective, but I feel like there is a clear advantage to uniform sampling over leverage score sampling, and I am not too concerned about those concerns raised by reviewers. However, I am not sure I agree with "however, we want to emphasize that our contributions address a fundamental assumption in spectral sparsification for graphs that has stood since the seminal paper of Srivastava and Spielman [3]" though, since uniform sampling is not a new idea and has been studied extensively.
> >
> > To me, my main concern is the one also discussed in the rebuttal to 51ez's review. "The fact that Uniform sampling works seems to heavily contradict conventional intuition. " I am not sure that I fully agree with this sentiment. On my first read through the paper, it seemed like the authors were saying something that is pretty intuitive. If I decompose my graph into well connected components (e.g. expanders), then uniform sampling on each of these components automatically gives us a spectral sparsifier, since leverage scores on expanders (read: well connected components) are uniform. What the authors prove is a slightly different statement, since they preserve the eigenspace of the whole graph, not just each component, but the overall result is not terribly surprising.

---

> > > ### Author Response · Authors · 2025-08-07
> > >
> > > We sincerely thank the reviewer for their thoughtful engagement and their kind words on the technical interest of the work.  We agree that our current phrasing — that our result “addresses a fundamental assumption in spectral sparsification since Srivastava and Spielman [3]” — may come across as too strong or overstated. Our intention was to highlight that, historically, importance sampling  (e.g., leverage scores or effective resistances) has been considered essential for spectral sparsification, while our contribution lies in identifying sufficient conditions in the form of structural assumptions  (i.e., large $\Upsilon(k)$) under which in fact uniform sampling is provably sufficient.
> > >
> > > Moreover, the prior works generally aim  to preserve the whole spectrum of the Laplacian, while our goal is more specific — to preserve just the bottom $k$ eigenspace, which aligns more closely with the needs of spectral clustering. Given the reviewer’s presented perspective, the results may feel intuitive, and we will revise to modulate our language accordingly. To clarify when we say conventional intuition, we are referring to analysis and the viewpoint through full effective resistances that Spielman and Srivasta and many others have studied. In the intuitive example that you provided, it is indeed simple to uniformly sample to sparsify when we consider just the well connected components. We believe that the unituitive aspect, and what causes difficulty in the analysis of that example, come from the cross cluster edges. In well clustered applications, the effective resistances of cross cluster edges are potentially extremely large (with dependence on $n$, resulting in vacuous bounds), which would dominate the importance when it comes to sampling compared to the intracluster edges. This unboundedness of the full effective resistance implies that uniform sampling will not preserve clustering through *conventional* analysis. The leap beyond intuition came from understanding what different eigenmodes correspond to in clustered graphs, and what eigenspace uniform sampling will preserve through the novel notion of rank $n-k$ effective resistances. We agree with the reviewer that uniform sampling has been heavily analyzed in other structured settings, and we greatly thank the reviewer for the valuable discussion in clarifying and repositioning the contributions of our paper within the larger literature.

---

> > > > ### Comment · Reviewer_R6LN · 2025-08-09
> > > >
> > > > I completely agree with what is said here! I think this line in particular highlights the issue this paper is trying to address, and is perhaps discussing the main issue I have with this paper that I was not able to properly articulate before. I think I did not do a good enough job during the initial review process, or was not able to fully articulate how I felt about this work.
> > > >
> > > > > In well clustered applications, the effective resistances of cross cluster edges are potentially extremely large (with dependence on , resulting in vacuous bounds), which would dominate the importance when it comes to sampling compared to the intracluster edges.
> > > >
> > > > I fully agree with what is said here. It is certainly true that uniform sampling would not work if we were trying to obtain spectral sparsifiers of the entire graph, and that this is precisely because the cross cluster edges can be high effective resistance. I think the authors also agree that if we were to want to sparsify each individual well connected component, then this is also simple as uniform sampling would suffice. The real question is the regime in between, we know that we can sparsify each individual well connected component if we do uniform sampling, but the whole graph will not be sparsified. What global properties are preserved exactly if we sample uniformly at random? This is the question the authors are trying to answer.
> > > >
> > > > In fact, there is a well known folklore algorithm to construct spectral sparsifiers that is somewhat related that comes to mind here. Instead of sampling edges proportional to their leverage scores, we can instead flip coins for all the edges with leverage score $< n/2m$, for which a constant fraction of the graph satisfies. Repeating this log n times gives an algorithm that takes $\widetilde{O}(m)$ time.
> > > >
> > > > I mention this because this seems similar in spirit to trying to sample just the edges that are connected components uniformly at random. I understand this is not what the authors are doing, but I am mentioning it nevertheless. These are precisely the edges that have low leverage score. The main issue with this approach is that there can be many other low leverage score edges that can be across cuts.
> > > >
> > > > I think the discussion above better elucidates how I felt initially about the paper. There are ways to perform spectral sparsification, if the graph is structured a certain way, say, very uninterestingly, connected clusters, with O(n) total edges across clusters. If there is no structure, or a lot, this is a very simple problem, but finding out the exact graph classes for which uniform sampling would work on is an interesting problem. Correspondingly, finding out what specific eigenspaces are preserved under uniform sampling is also an interesting problem.
> > > >
> > > > This is why my initial questions focused so much so on what this class of graphs the authors were discussing looked like. To me, the authors initial response describing this class of graphs was the most interesting part, because it describes exactly a class of graphs for which uniform sampling would work.
> > > >
> > > > However, the main reason why I did not give a higher score, is because it does not seem obvious to me what the applications of this work would be. I might be completely wrong, but I do not really see where these bounds would find use in.

---

> > > > > ### Author Response · Authors · 2025-08-09
> > > > >
> > > > > We thank the reviewer for acknowledging the algorithmic contribution. Our results fit within the emerging paradigm of \emph{structure-driven speedups} (e.g., Huang/Vishnoi, Braverman et al). Further, our results are accompanied by analysis tools that we believe are of independent interest including a novel rank-$(n-k)$ effective resistance notion (that isolates the subspace orthogonal to the cluster-indicator span) and an eigenspace-specific matrix concentration technique (a matrix Chernoff bound tailored to the top $(n-k)$-dimensional eigenspace). On the application side, the guarantees enable scalable pipelines in settings where preserving specific Laplacian eigenvectors is algorithmically critical,such as graph-based semi-supervised learning (label propagation on $L$), computation of spectral embeddings for large networks, and dynamic sparsification where one seeks to maintain a structure-preserving sketch under edge updates. In summary, our contributions close a concrete gap by proving that uniform sampling suffices under standard clusterability assumptions while yielding principled runtime improvements for spectral clustering and related downstream spectral methods.

---

### Official Review · Reviewer_irHX · 2025-07-02

**Clarity:** 3
**Significance:** 3
**Originality:** 3
**Rating:** 4
**Confidence:** 3

**Summary:**

This paper investigates whether uniform edge sampling can suffice for spectral clustering in graphs with well-separated clusters. The main contribution is a theoretical analysis showing that uniform sampling preserves the spectral subspace necessary for clustering under strong clusterability conditions. The authors derive resistance bounds for intra-cluster edges to analyze uniform sampling for clustered graphs.

**Questions:**

If I understand correctly, one main practical conclusion of this work is to show that the expensive effective resistance sampling does not surpass trivial uniform sampling as much as we'd expect. What about other sampling strategies?

**Ethical Concerns:**

["NO or VERY MINOR ethics concerns only"]

**Final Justification:**

The primary contributions of this work are theoretical in nature, and the overall quality is robust, albeit constrained by certain simplifying assumptions necessary to facilitate the proof.

**Limitations:**

yes

**Paper Formatting Concerns:**

Cross-reference fails on lines 210 and 217

**Quality:**

3

**Strengths And Weaknesses:**

Strengths
1. Understanding the side effects of uniform edge sampling is an important topic for graph sparsification. The paper provides a novel theoretical analysis showing that uniform sampling suffices for spectral clustering under strong clusterability conditions.
2. The results suggest that simple and efficient uniform sampling can be effective for large-scale spectral clustering.
3. This paper introduces resistance bounds and the matrix Chernoff bound, which may be of independent interest for spectral graph theory.

Weaknesses
1. While the theory appears compelling, the numerical experiments are limited to small random graph models.
2. Real-world graphs are seldom well-clustered so the structure ratio $\Upsilon(k)$ is usually not large enough for the discussions to hold.

---

> ### Author Rebuttal · Authors · 2025-07-30
>
> > While the theory appears compelling, the numerical experiments are limited to small random graph models.
>
> We would like to emphasize that this work is, primarily, a theoretical contribution that establishes a new understanding of sparsification and provides new analytical tools, following the practice of past seminal papers in spectral graph theory. We use the flexible SBM experiments to validate our theory, consistent with what has also been done in previous works, e.g., [1,3]. This allows us to have precise control over graph constructions to verify our theoretical contributions. Further development of other algorithms as suggested by the reviewer are indeed important future directions which will require deeper empirical evaluations.
>
> > Real-world graphs are seldom well-clustered so the structure ratio is usually not large enough for the discussions to hold.
>
> We acknowledge that not all real graphs for clustering satisfy our conditions, and our results will not be applicable for those graphs. Our contributions use clusterability assumptions that have been established in prior seminal works like Peng et al. and Macgregor et al. [1,3] . We want to emphasize that our contribution includes new tools, such as the rank (n-k) effective resistance, to inform practical algorithm design.
>
> > If I understand correctly, one main practical conclusion of this work is to show that the expensive effective resistance sampling does not surpass trivial uniform sampling as much as we'd expect. What about other sampling strategies?
>
> On one hand, we have the optimal sampling regime of effective resistance sampling, and on the other hand, we have oblivious uniform sampling, which we demonstrate is sufficient for sampling. A very promising future direction is to use a uniform subsample to compute a coarse approximation of the effective resistance. We can subsequently sample using the approximate effective resistances. This technique has been explored in Cohen et al. [4] and can be extended to our setting. We will discuss this in our revision.
>
> [1] Peter Macgregor and He Sun. A tighter analysis of spectral clustering, and beyond.
>
> [2] Daniel A. Spielman and Nikhil Srivastava. Graph sparsification by effective resistances.
>
> [3] Richard Peng, He Sun, and Luca Zanetti. Partitioning well-clustered graphs: Spectral clustering works!
>
> [4] Michael B. Cohen, Yin Tat Lee, Cameron Musco, Christopher Musco, Richard Peng, and Aaron Sidford. Uniform sampling for matrix approximation.

---

> > ### Comment · Reviewer_irHX · 2025-08-03
> >
> > Dear authors, thank you for your response. I acknowledge the theoretical contributions and the controlled assumptions necessary for the theoretical analyses. I have no further questions and maintain my original rating.

---

### Official Review · Reviewer_UEcv · 2025-07-02

**Clarity:** 3
**Significance:** 3
**Originality:** 3
**Rating:** 5
**Confidence:** 3

**Summary:**

The paper investigates the effectiveness of uniform edge sampling for spectral sparsification in the context of spectral clustering. Traditionally,  spectral sparsification involves the approximate computation of effective resistance values for sampling. This work, however, demonstrates that under certain graph-structural assumptions, specifically quantified by a structure ratio $\Upsilon$, simple uniform sampling can effectively preserve spectral clustering structure. High structure ratio $\Upsilon(k)$ implies that the clusters within the graph are well-separated and the graph has a clear $k$ - partitioning structure. Under such conditions, the authors show that uniform edge sampling (randomly selecting edges without bias) sufficiently preserves spectral clustering quality, eliminating the need for other importance-based sampling methods.

The main theoretical result establishes that uniformly sampling edges preserves the eigenspaces crucial to clustering, given a sufficiently large structure ratio and well-conditioned rank properties. Experiments using stochastic block model (SBM) graphs confirm that uniform sampling performs comparably to more expensive sampling strategies under these conditions.

**Questions:**

See the weakness section.

**Ethical Concerns:**

["NO or VERY MINOR ethics concerns only"]

**Final Justification:**

This work’s main contributions are theoretical, and the overall quality is strong. I maintain my score.

**Limitations:**

yes

**Paper Formatting Concerns:**

No major formatting issues.

**Quality:**

4

**Strengths And Weaknesses:**

Strengths:
1) The paper identifies conditions under which uniform sampling suffices for spectral sparsification.

2) Introduces rank‑(n‑k) effective‑resistance formulation and subspace‑specific Chernoff bound.

The paper shows that if a graph is genuinely clusterable, expensive importance sampling is unnecessary. This is a non‑trivial theoretical contribution that sharpens our understanding of when uniform sampling works. While the practical section is thin, the conceptual result fills a gap between coreset theory and spectral sparsification and should interest both theoretical and applied communities.

Weakness:

1) Evaluating the method on graph families beyond SBMs would better demonstrate its empirical robustness.

2) The experiments could include synthetic graphs that span a range of condition numbers.

Related work:

Suggestion: include the following work in the related work section. Efficient approximation algorithms exist to estimate effective resistance. However, that doesn’t lessen the significance of the results established in this paper.

Local algorithms for estimating effective resistance, KDD 2021.

---

> ### Author Rebuttal · Authors · 2025-07-30
>
> > Evaluating the method on graph families beyond SBMs would better demonstrate its empirical robustness
>
> Thank you for this suggestion. We agree that evaluating our approach on additional graph families would strengthen the empirical validation. In a revision, we would include experiments on other well-studied graph models, such as Degree-Adjusted SBMs and also Hierarchical Random Graphs to evaluate the performance of uniform sampling under finer-grain cluster structures. We will also try larger-scale experiments to see if there are any significant changes. Preliminary experiments on the above families during the rebuttal period seem to indicate that our approach still works in those families of graphs, and in larger-scale graphs, as predicted by our theoretical bounds.
>
> > The experiments could include synthetic graphs that span a range of condition numbers.
>
> This is an excellent suggestion! We will include experiments that systematically vary the condition number and structure ratio $\Upsilon(k)$ to empirically validate how these parameters affect uniform sampling performance, providing clearer practical guidance on when our method is most effective. In particular, we also need to understand the connection between the condition number of, say, SBM-generated graphs and the clustering parameter $\Upsilon(k)$. In preliminary evaluations and in order to control the condition number, we had to vary the cluster sizes and we observed, as expected, a correlation between the ratio of the largest to the smallest cluster and the condition number. In preliminary evaluations, this does not significantly affect the performance of our approach, but more extensive experiments are needed.
>
> > Suggestion: include the following work in the related work section. Efficient approximation algorithms exist to estimate effective resistance. However, that doesn’t lessen the significance of the results established in this paper. Local algorithms for estimating effective resistance, KDD 2021.
>
> Thank you for the reference to the KDD 2021 work. Indeed, there exist many algorithms that seek to approximate effective resistances both theoretically and empirically. We will add this discussion to our related work section. It is worth noting that all these algorithms only guarantee additive error approximations: our bounds when exact effective resistances are used to sample edges would still hold if relative-error approximations or constant factor approximations to the effective resistances were used (with a small loss in the quality of our guarantees). However, this is not the case when additive error approximations to the effective resistances are used.

---

> > ### Comment · Reviewer_UEcv · 2025-08-02
> >
> > Thank you for your detailed rebuttal. I will keep my score.

---

### Note · Authors · 2025-08-12

We greatly thank all of the reviewers and area chair for their time and effort. To summarize, we introduce two novel contributions in the area of spectral graph sparsification and spectral clustering.

First, we show that by using effective resistances, we can sparsify the input graph using $O(n \log n)$ samples and obtain clustering guarantees comparable to clustering the original graph. Second, under standard clusterability assumptions, we show that uniform sampling suffices, and we obtain sampling bounds of $O(\gamma ^2 n\log n )$, where $\gamma$ is a structural term that is **independent** of the size of the graph. This is the first theoretical guarantee that uniform sampling edges preserve spectral clustering structure under clusterability.

Our analysis hinges on a novel theoretical insight -- we need to preserve only the bottom-$k$ eigenspace for clustering instead of the full spectrum considered in prior works. We also introduce new tools -- novel rank-$(n-k)$ effective resistance bounds, and specialized matrix concentration analyses tailored for clustering-relevant eigenspaces, which may be of independent interest.

Our experiments demonstrate that uniform sampling achieves subspace approximation quality matching effective resistance sampling on well-clustered graphs, validating our theoretical predictions. Regarding run-time comparisons with approximate effective resistance methods: reviewers pointed out numerous methods for approximating effective resistances (which we will cite and make a note in the experiments section); however, as another reviewer noted, such evaluations are beyond our scope, and uniform sampling remains fundamentally simpler and more efficient, requiring no preprocessing while achieving comparable spectral guarantees under our structural assumptions.

---

### Decision · Program_Chairs · 2025-09-17

**Decision:**

Accept (poster)

**Comment:**

The authors present a new algorithm for spectral sparsification based on uniform sampling. While in previous work uniform sampling is used as a step to compute good sparsifiers, it is interesting to obtain good bound to get spectral sparsifier via uniform sampling.

The reviewers found the paper interesting and techniques nice although they all felt that the paper would improve significantly by:
- adding a more in depth comparison with the worked based on effective resistance
- adding a detailed comparison with the work "Uniform Sampling for Matrix Approximation"
- the experimental analysis is very limited and it would be nice to have a more comprehensive list of experiments